



Atmospheric
Measurement
Techniques

# Boundary layer water vapour statistics from high-spatial-resolution spaceborne imaging spectroscopy

**Mark T. Richardson**[1,2]**, David R. Thompson**[1]**, Marcin J. Kurowski**[1]**, and Matthew D. Lebsock**[1]

[1]Jet Propulsion Laboratory, California Institute of Technology, Pasadena, CA 91109, USA
[2]Department of Atmospheric Science, Colorado State University, Fort Collins, CO 90095, USA

**Correspondence:** Mark T. Richardson (markr@jpl.nasa.gov)

**Abstract.** Daytime clear-sky total column water vapour (TCWV) is commonly retrieved from visible and shortwave infrared reflectance (VSWIR) measurements, and modern missions such as the upcoming Earth Surface Mineral Dust Source Investigation (EMIT) offer unprecedented horizontal resolution of order 30–80 m. We provide evidence that for convective planetary boundary layers (PBLs), spatial variability in TCWV corresponds to variability in PBL water vapour. Using an observing system simulation experiment (OSSE) applied to large eddy simulation (LES) output, we show that EMIT can retrieve horizontal variability in PBL water vapour, provided that the domain surface is uniformly composed of either vegetated surfaces or mineral surfaces. Random retrieval errors are easily quantified and removed, but biases from $-7\%$ to $+34\%$ remain in retrieved spatial standard deviation and are primarily related to the retrieval's assumed atmospheric profiles. Future retrieval development could greatly mitigate these errors. Finally, we account for changing solar zenith angle (SZA) from 15 to 60° and show that the non-vertical solar path destroys the correspondence between footprint-retrieved TCWV and the true TCWV directly above that footprint. Even at the 250 m horizontal resolution regularly obtained by current sensors, the derived maps correspond poorly to true TCWV at the pixel scale, with $r^2 < 0.6$ at SZA = 30°. However, the derived histograms of TCWV in an area are closely related to the true histograms of TCWV at the nominal footprint resolution. Upcoming VSWIR instruments, primarily targeting surface properties, can therefore offer new information on PBL water vapour spatial statistics to the atmospheric community.

*Copyright statement.* Jet Propulsion Laboratory, California Institute of Technology. Government sponsorship acknowledged.

## 1 Introduction

Thermodynamic information about the planetary boundary layer (PBL), including information about water vapour ($q_v$), is a targeted observable recommended by NASA's Decadal Survey (National Academies of Science, Engineering, and Medicine, 2018TS1). PBL $q_v$ estimates would go beyond the current total column water vapour (TCWV) and free-tropopause products to provide new information about the vertical moisture structure for weather and climate applications. The Decadal Survey explicitly recognised the PBL's importance since it "literally couples the surface of the Earth to the atmosphere above", and among other important factors, gradients of moisture between the surface and PBL and between the PBL and free troposphere are strong controls on vertical atmospheric heat and moisture transport. The formation of boundary layer clouds was also highlighted due to their importance for Earth's energy balance. A critical measurement gap in the current observations of PBL thermodynamics is the inability to quantify mesoscale variations in PBL $q_v$. Mesoscale aggregation in PBL water vapour appears to play an important role in determining the timing of deep convective events (Stirling and Petch, 2004; Wulfmeyer et al., 2006). Furthermore, in situ observations suggest that the majority of the variation in the TCWV prior to convective initiation can be explained by variability within the PBL (Couvreux et al., 2009). The mesoscale spatial variability of $q_v$ is not resolved by current global weather or climate models, but instead it must be parameterised. Modern approaches to

Please note the remarks at the end of the manuscript.

parameterise PBL variability include eddy-diffusivity/mass-flux approaches (Suselj et al., 2019) and higher-order closure approaches that include prognostic equations for higher-order moments such as the variance (Golaz et al., 2002; Larson et al., 2002). However, we lack observations at a global scale to evaluate the small-scale variability produced by these models. This paper will address the feasibility of addressing this measurement gap using upcoming observations from very high-spatial-resolution visible and shortwave infrared reflectance (VSWIR) observations from space.

This study is primarily motivated by the ongoing development of spaceborne hyperspectral VSWIR measurement capacity at fine horizontal resolution. We focus on the EMIT mission, planned to launch to the International Space Station (ISS) in 2022 with an average footprint size ($\Delta x$) of 60 m (Green and Thompson, 2020). However, similar or improved capacity is anticipated in response to NASA's Surface Biology and Geology (SBG) designated observable, with the Hyperspectral Infrared Imager (HysPIRI, Lee et al., 2015) concept offering $\Delta x$ of 30–60 m, and for ESA's Copernicus Hyperspectral Imaging Mission for the Environment (CHIME), also known as Sentinel 10, for which the prime contractor was selected in July 2020 and whose Mission Requirements Document refers repeatedly to $\Delta x < 30$ m (Rast et al., 2019).

Of present missions, this analysis may be applicable to the Italian PRecursore IperSpettrale della Missione Applicativa (PRISMA, Candela et al., 2016), which provides similar spectral range and sampling to EMIT at $\Delta x = 30$ m. Some of the conclusions will also apply to other recent instruments, such as Sentinel-2's Multi-Spectral Imager (MSI, Drusch et al., 2012), which offers $\Delta x = 20$ m, albeit with far fewer channels, or the DLR Earth Sensing Imaging Spectrometer (DESIS, Krutz et al., 2019), which provides hyperspectral measurements over a smaller wavelength range. These modern and upcoming instruments offer $\Delta x$ that are substantially smaller than past VSWIR instruments that retrieve TCWV, such as ESA's Medium Resolution Imaging Spectrometer (MERIS) on Envisat, whose smallest provided $\Delta x$ is approximately 0.25 km × 0.30 km, which allowed the identification of horizontal convective rolls during a high-pressure event over Germany (Carbajal Henken et al., 2015) but cannot resolve the smaller scales of variability. Recently, Thompson et al. (2021) used VSWIR measurements from the Airborne Visible Infrared Imaging Spectrometer-Next Generation (AVIRIS-NG) to capture information about PBL $q_v$ variability at spatial scales < 1 km, which cannot be determined with footprint sizes similar to MERIS.

EMIT-like instruments could allow retrieval of bulk PBL $q_v$, which we henceforth refer to as the partial column water vapour in the PBL (PCWV$_{PBL}$) via two demonstrated approaches. The first approach uses VSWIR measurements alone, and the second combines separate above-PBL water vapour (PCWV$_{upper}$) and TCWV to obtain PCWV$_{PBL}$ = TCWV − PCWV$_{upper}$. A third approach, which has not been demonstrated operationally to our knowledge, is to perform joint retrievals using both VSWIR and vertically resolved sounding measurements.

The direct VSWIR-only method can be seen in Trent et al. (2018), who estimated PCWV$_{PBL}$ from the Greenhouse Gases Observing Satellite (GOSAT, Kuze et al., 2009), while the second is explored in Millán et al. (2016), who paired TCWV from passive microwave measurements with PCWV$_{upper}$ above horizontally uniform clouds from Moderate Resolution Imaging Spectroradiometer (MODIS) near-infrared retrievals. The resultant PCWV$_{PBL}$ values showed good agreement with radiosondes and ERA-Interim reanalysis, and a promising candidate approach is to use VSWIR TCWV in place of the microwave measurements.

The physical principle of VSWIR TCWV retrievals is differential optical absorption spectroscopy (DOAS). More TCWV leads to increasing depth of $H_2O$ absorption features relative to other wavelengths. This applies to TCWV$_{VSWIR}$ from missions including MERIS (Bennartz and Fischer, 2001; Guanter et al., 2008), MODIS (Diedrich et al., 2015; Gao and Kaufman, 2003), TROPOMI (Borger et al., 2020; Schneider et al., 2020), SCIAMACHY (Noël et al., 2004), GOME (Noël et al., 1999), GOME-2 (Grossi et al., 2015) and OCO-2 (Nelson et al., 2016).

These instruments vary in spectral range and sampling, but all must contend with the measured spectra responding to properties other than TCWV. The retrievals only operate for daytime cloud-free scenes and commonly only over land, since water surfaces are dark such that insufficient light reaches the sensor to allow for a TCWV retrieval, with exceptions for sun glint as exploited in the aforementioned AVIRIS-NG study (Thompson et al., 2021). Thompson et al. selected these AVIRIS-NG flights because DOAS techniques respond to the total light path absorption including the slanted sunlight path from the top of atmosphere (TOA) to the surface. This horizontally smears the effective footprint size, with larger smearing for larger solar zenith angle (SZA). As footprints become smaller, the proportional effect of this smearing may become more important, and so here we apply solar ray tracing to determine whether observations with a nominal $\Delta x$ of 20–50 m obtain useful information about the spatial statistics of PCWV$_{PBL}$ at that spatial resolution. We use two performance metrics: (i) the correlation between retrieved TCWV and true TCWV, which was used as input for our forward simulations, and (ii) the spatial standard deviation $\sigma_x$ of retrieved TCWV within a snapshot relative to the large eddy simulation (LES) output PCWV$_{PBL}$ $\sigma_x$, which we refer to as the true $\sigma_x$.

We employ a new type of Observing System Simulation Experiment framework and perform simulated VSWIR retrievals of TCWV from high-spatial-resolution LES output to determine whether horizontal spatial variability in PBL $q_v$ can be obtained from retrieved TCWV, and conclusions are limited to daytime non-cloudy conditions. The purpose of this is a detailed sensitivity study using retrieval code and tools already developed for EMIT. We consider $\Delta x \geq 40$ m

since this is appropriate for EMIT and several LES cases in our archive were run at that resolution.

Here we test the use of the iterative optimal estimation code Imaging Spectrometer Optical Fitting (Isofit) for a spaceborne application, specifically target TCWV and address the following questions.

1. In LES, how does horizontal variability in TCWV relate to PCWV$_{PBL}$?

2. What uncertainties are introduced into the retrieval by EMIT instrumental error, non-uniform AOD and different surface types, and can these errors be anticipated and quantified from observations alone?

3. What is the correlation coefficient between retrieved and true TCWV, and can the spatial standard deviation be estimated? How does this depend on LESs of different convective PBL types?

4. How does the solar path across different SZAs affect these conclusions?

This scope excludes important factors such as topography, inter-channel correlated errors in the instrument, imperfect cloud masking and cloud 3D radiative effects, and our paper is structured to address these questions in turn, with each analysis section containing its own methodology, results and discussion. Section 2 explores the raw LES output to address question 1, Sect. 2 describes the synthetic retrievals and analysis methodology to address questions 2–3, Sect. 4 adds solar path analysis to address question 4, and Sect. 5 discusses and concludes.

## 2   Large eddy simulations

### 2.1   Model setup, scenarios and snapshot selection

We use output from five LES simulations named RICO, ARM, ARM_lsconv, BOMEX and DRY, which are summarised with references in Table 1. They all represent convective boundary layers characterised by either low-altitude or no cloud cover. The 23 separate snapshots are identified by timestamp; e.g. ARM_18000s is 5 h into the ARM simulation. Simulation $\Delta x$ sets the implied measurement horizontal resolution and varies from 20 to 50 m.

The simulations are performed with two different models: EULAG (Prusa et al., 2008; ARM and ARM_lsconv) and JPL-UCONN LES (Matheou and Chung, 2014; RICO, BOMEX, DRY). Each simulation applies periodic lateral boundary conditions and a horizontally homogeneous initial state. For the RICO case, interactive sensible and latent heat surface fluxes over constant-temperature ocean are used, while the other cases are driven by prescribed (either constant for DRY and BOMEX or time-dependent for ARM and

ARM_lsconv) surface fluxes. All other setup details are explained in the Table 1 references: these references also show how the ARM, BOMEX and RICO LES simulations, which were based on detailed field campaigns, accurately reproduce the main features observed during those campaigns. Each 3D LES snapshot is merged with 1D MERRA-2 reanalysis profiles aloft to produce a full-depth atmospheric column. Reanalysis data are chosen for the dates and locations of the field campaigns the LESs refer to. The DRY and ARM_lsconv cases share the same upper-atmospheric profiles as ARM. In all cases except for DRY, Table 1 rows (vii)–(ix) show that the LESs capture > 85 % of total TCWV. For retrieval purposes we ignore the LES surface type and apply an assumed surface reflectance spectrum below the LES profiles.

### 2.2   Profiles and PBL height

Definitions of PBL height $z_{PBL}$ vary widely. We found similar results from four standard thermodynamic calculations (von Engeln and Teixeira, 2013), so henceforth we define $z_{PBL}$ as the altitude of $\max(d\theta/dz)$, where $\theta$ is the all-sky mean potential temperature. Mean all-sky profiles of $T$ and $q$, horizontal standard deviation in $q$ ($\sigma_q$), and cloud fraction are shown in Fig. 1. Changes in $\sigma_q$ are the largest differences between time steps but are small ($< 10\,\%$) relative to the mean, so measuring this variability will require precise observations. Also, $\sigma_q$ is negligible in the layers in the free troposphere that lie above the PBL but are resolved by the LES, implying that the LES domains capture $q_v$ variability. We later support this claim using real-world airborne lidar retrievals.

### 2.3   Water vapour spatial variability statistics and the relationship between TCWV and PCWV$_{PBL}$

Figure 1 displays all-sky conditions, but our retrievals only target clear sky, thereby missing a moister tail to the distribution (Supplement Fig. 1). Within-cloud retrievals would require alternative measurement approaches, such as differential absorption radar (Roy et al., 2018, 2020), and the restriction to clear-sky scenes is a limitation that also applies to current thermal infrared and lidar retrievals.

We assess TCWV-PCWV$_{PBL}$ spatial variability by calculating clear-sky PCWV up to capping altitudes from 0.5 to 5 km and then correlating these with TCWV. Figure 2 confirms that > 90 % of horizontal variance in LES TCWV at these scales is explained by PCWV$_{PBL}$. It is reasonable to ask whether this finding that the PBL variance dominates the TCWV variance is representative of the real atmosphere. Indeed, the LES results are supported by the same statistics calculated from High Altitude Lidar Observatory (HALO) flights over the Pacific Ocean in April 2019 (Bedka et al., 2021), as presented in Thompson et al. (2021) and shown in Fig. 2f. In these calculations TCWV is only calculated up

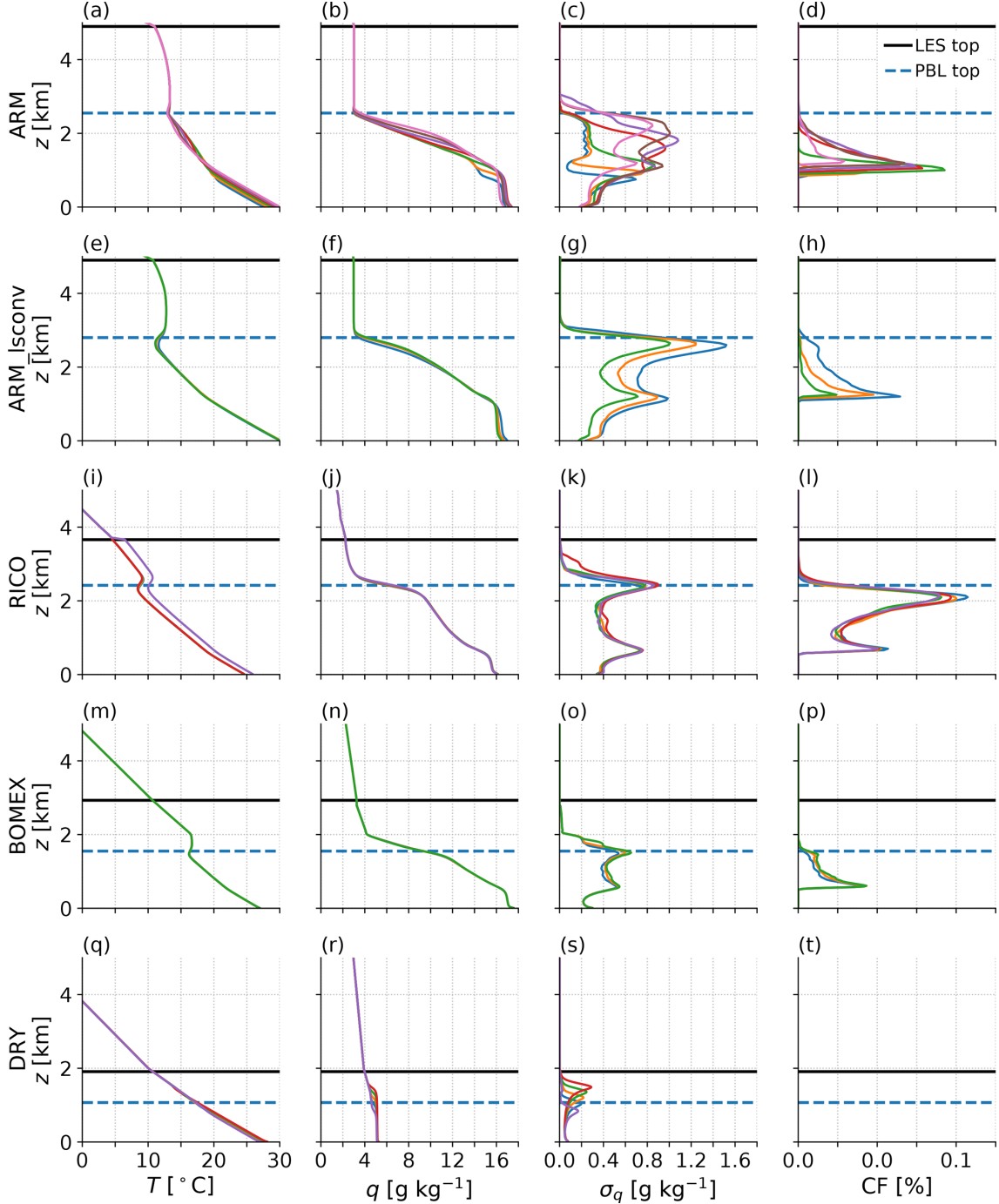

**Figure 1.** Output all-sky profiles for the **(a–d)** ARM, **(e–h)** ARM_lsconv, **(i–l)** RICO, **(m–p)** BOMEX and **(q–t)** DRY LES. In each panel the separate coloured lines represent different timesteps, the black horizontal line is the top of the LES and the dashed blue horizontal line is the PBL height calculated from the first shown timestep, whose lines are in the same blue. The column beginning with **(a)** is the mean $T$ profile, that with **(b)** the mean $q$ profile, that with **(c)** the profile of the spatial standard deviation in $q$ and that with **(d)** the cloud fraction. Note that due to overlap, the fraction of cloudy columns listed in Table 1 is higher than the peak mean profile cloud fraction.

**Table 1.** Summary of LES properties. Where ranges are provided, these are the full range of clear-sky mean values from the snapshots used for each case. Row (vi) is the fraction of columns whose integrated liquid water path $> 1.3 \times 10^{-3}$ mm and differs from mean cloud fraction in Fig. 1 due to overlap. The TCWV in row (vi) is derived from the combined LES and reanalysis profile and separated into the LES and reanalysis partial column water vapour amounts in rows (vii) and (viii).

| | | ARM | ARM_lsconv | RICO | BOMEX | DRY |
|---|---|---|---|---|---|---|
| (i) | Snapshots used | 18000s, 21600s, 25200s, 28800s, 32400s, 36000s 43200s | 36000s, 39600s, 43200s | 14400s, 16200s, 18000s, 19800s, 21600s | 14400s, 16200s, 18800s | 7200s, 10800s, 14400s, 18000s, 21600s |
| (ii) | Domain size (km) | 20.0 | 20.0 | 20.5 | 12.8 | 14.4 |
| (iii) | $\Delta x$ (m) | 50 | 50 | 40 | 20 | 20 |
| (iv) | LES top (km) | 5 | 5 | 4 | 3 | 2 |
| (v) | PBL top (km) | 1–2.7 | 1–3.2 | 2.5–2.7 | 2.1 | 1.3 |
| (vi) | Columns flagged cloudy (%) | 1–21 | 5–20 | 24–28 | 16–19 | 0.0 TS2 |
| (vii) | Clear-sky TCWV (mm) | 39.6–42.2 | 43.3–43.8 | 36.9–37.0 | 35.6–35.7 | 19.8–20.2 |
| (viii) | Clear-sky PCWV$_{LES}$ (mm) | 36.2–38.9 | 40–40.5 | 33.1 | 30.6–30.7 | 9.7–10.2 |
| (ix) | Clear-sky PCWV$_{reanalysis}$ (mm) | 3.3 | 3.3 | 3.9 | 5.0 | 9.9 |
| (x) | Description | Diurnal cycle of midlatitude shallow convection over land | As ARM, perturbed by large-scale convergence | Shallow precipitating trade-wind convection over ocean | Shallow non-precipitating trade-wind convection over ocean | Dry free convection |
| (xi) | Citation | Brown et al. (2002). REF case in Kurowski et al. (2020) | CON3 case in Kurowski et al. (2020) | vanZanten et al. (2011); Matheou and Chung (2014) | Siebesma et al. (2003); Matheou and Chung (2014) | Matheou and Chung (2014) |

to 8 km due to flight altitude, but these real-world data include free-tropospheric moisture variability and furthermore will have lower $r$ values due to the presence of random retrieval error. The horizontal resolution is $\sim 3$ km versus the 20–50 m of LES, and the HALO sampling is sparse and often separated by hundreds of kilometres due to clouds. Nevertheless, the HALO flights show that horizontal TCWV variability can be well captured within 3 km altitude in real scenes and provide evidence that the LES domains capture horizontal variability in $q_v$.

The TCWV-PCWV$_{PBL}$ fit coefficients for ARM, ARM_lsconv, BOMEX and RICO range from 0.99 to 1.04 mm mm$^{-1}$; i.e. a 1 mm change in PCWV$_{PBL}$ means a 0.99–1.04 mm change in TCWV. This confirms that almost all horizontal $q_v$ variability occurs within the mean PBL height. For the DRY case, coefficients range from 1.06 to 1.12 mm mm$^{-1}$. These coefficients mean that PCWV$_{upper}$ spatially correlates with PCWV$_{PBL}$, which could be explained by moister plumes rising and having higher local $z_{PBL}$ than the domain-mean value used in the calculation.

In summary, we have answered question 1 from Sect. 1 and can expect spatial variability in retrieved TCWV for these cases to represent real variability in PCWV$_{PBL}$, and so we use TCWV and PCWV$_{PBL}$ interchangeably from now on.

## 3  Simulated EMIT retrievals of TCWV in LES

This experiment requires a large number of inversions over a wide spatial field. Simulating synthetic spectra and performing a retrieval for every grid point proved to be prohibitively computationally expensive. Consequently, we develop an emulator to statistically reproduce the result of the full inversion but with dramatically better efficiency. Retrievals will include a range of surfaces in a subset of the snapshots (to identify sensitivity to surface type) and then a fixed surface type across all snapshots (to identify sensitivity to atmospheric conditions). Sensitivity tests will be performed on individual subsets of snapshots as required, and a correction method for identifying the random component

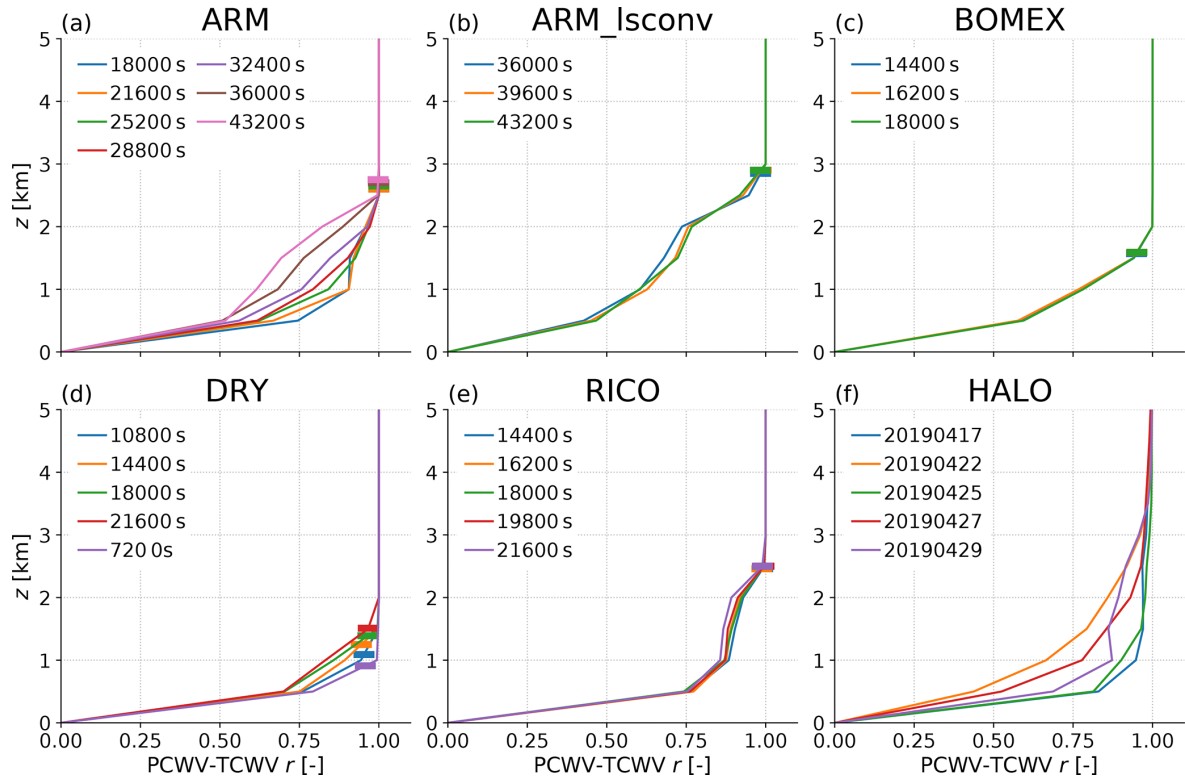

**Figure 2.** Correlation coefficient between clear-sky partial column water vapour (PCWV) integrated up to given capping altitudes, and the TCWV. **(a)**–**(e)** contain the snapshots of each individual LES run and **(f)** reproduces the values calculated from High Altitude Lidar Observatory flights over the Pacific (Bedka et al., 2021) as presented in Thompson et al. (2021). The LES profiles also have a horizontal bar appended at the derived PBL top height. The flight data differ from the LES outputs in that horizontal resolution is approximately 3 km along-track, they are dispersed over thousands of kilometres, and the TCWV is only up to 8 km due to the flight altitudes.

of retrieval error will be introduced. Section 3.1 describes the relevant methods, Sect. 3.2 gives the results and Sect. 3.3 discusses limitations.

### 3.1 Retrieval methodology

#### 3.1.1 MODTRAN6.0 forward model, EMIT instrument characteristics and Isofit retrievals

We use the same retrieval code as in Thompson et al. (2021), Imaging Spectrometer Optimal Fitting (Isofit), for our synthetic retrievals (https://github.com/isofit/isofit, last access: 2 June 2021 TS3). This iterative optimal estimation code simultaneously retrieves surface reflectance, aerosol optical depth (AOD) and TCWV, differing from older techniques that retrieve properties sequentially (e.g. Guanter et al., 2008 for MERIS). Isofit is described and shown to have a closed error budget in Thompson et al. (2018) and has been applied to observations from several airborne campaigns (Thompson et al., 2019, 2020, 2021). Conceptually it targets surface reflectance $\rho_s$, and the estimation of TCWV is seen as part of an atmospheric correction.

Forward simulations use MODTRAN6.0 (Berk et al., 2014, 2015), which provides a plane-parallel solution to the radiative transfer equation. Atmospheric reflectance and transmittance vectors $\rho_a$, $t$ and spherical sky albedo $s$ are calculated at wavenumber separation $\Delta k = 0.1\,\mathrm{cm}^{-1}$ ($\Delta\lambda \approx 0.002\,\mathrm{nm}$) before being convolved with the EMIT spectral response function, and the instrument is assumed to be nadir-viewing from 100 km altitude. With no substantial atmosphere above 100 km, this gives the same results as the ISS altitude near 400 km, where EMIT will be hosted. A correlated-$k$ method and the HITRAN database (Rothman et al., 2009) are used for gaseous absorption, while scattering is handled by DISORT (Laszlo et al., 2016; Stamnes et al., 1988). The EMIT instrument properties are derived from the current mission instrument model, which accounts for all signal-independent noise terms like electronic noise, and photon shot noise calculated using predicted efficiencies of the instrument mirrors, lens, grating, and focal plane array. The spectral range is 380–2500 nm, with $\Delta\lambda_{\mathrm{channel}} = 10\,\mathrm{nm}$ and full width at half maximum averaging $\Delta\lambda_{\mathrm{FWHM}} \approx 11\,\mathrm{nm}$.

For forward simulations, merged LES-reanalysis $T$ and $q$ profiles are interpolated onto a profile with eight points from 0 to 6 km, and then vertical resolution slowly degrades over 6–100 km. Interpolated TCWV differs from the LES reanal-

ysis, but we assume that conclusions regarding derived sensitivities and errors will not be strongly affected.

The forward radiance vector $I$ is calculated using a standard Lambertian approximation (e.g. as in Vermote et al., 1997):

$$I = \frac{I_0 \mu_0}{\pi} \left[ \rho_a + \frac{t \circ \rho_s}{1 - s \circ \rho_s} \right], \tag{1}$$

where $I_0$ is the downward TOA solar radiance, $\mu_0$ the cosine of the solar zenith angle, $\rho_s$ is the surface reflectance and $\circ$ represents channel-by-channel multiplication. The $\rho_s$ elements represent the hemispheric-directional distribution function (Schaepman-Strub et al., 2006). The atmospheric coefficient vectors $t$, $\rho_a$ and $s$ represent the transmittance of the solar-reflected optical path, the path reflectance, and the spherical sky albedo, respectively. These coefficients are obtained from simulations over a black surface. Using Eq. (1) in forward simulations results in negligible differences to retrieved TCWV compared with inserting the surface directly into MODTRAN forward simulations (Supplement Fig. 2). Use of Eq. (1) means that just one MODTRAN simulation is needed per column rather than one for each combination of column and surface type. The pseudo-observation, $I_{obs}$, is $I$ with random uncorrelated noise added, generated using the EMIT noise model.

The $I_{obs}$ are input as observations to Isofit, while its state vector $x$ contains surface reflectance in each channel, TCWV and aerosol optical depth at $\lambda = 550$ nm (AOD), i.e. $x = [\rho_s \text{ AOD TCWV}]$. We mask the most strongly absorbing channels due to lack of any surface information, so the retrieval uses 176 EMIT channels, and therefore $x$ has $176 + 2 = 178$ elements.

The $\rho_s$ elements are constrained via a covariance matrix whose mean is derived from a library of real surfaces, thereby capturing realistic spectral shapes. We retrieve absolute $\rho_s$, rather than the normalised value discussed in Thompson et al. (2018), and the prior is loosely constrained, however, ensuring that most information comes from the measurements.

Isofit uses Eq. (1) with a lookup table (LUT) for its forward model, populating $\rho_a$, $t$, and $s$ for selected AOD and TCWV and then linearly interpolating in TCWV, AOD space to estimate $I_{obs}$ given $x$. The LUT uses the default midlatitude summer profile and scales its $q(z)$ and aerosol extinction $(z)$ TS4 to match desired AOD (from 0.05 to 0.30) and TCWV (from 5 to 53.5 mm). The Isofit default configuration uses the U.S. Standard Atmosphere 1976 (Sissenwine et al., 1976), but MODTRAN applies a relative humidity limit, and the U.S. Standard Atmosphere 1976 is cool enough that MODTRAN automatically restricts its moisture content, such that the TCWV cannot reach the values seen in any LES case except for DRY. The midlatitude summer TCWV limit is just over 53.5 mm, so that defines our LUT maximum.

Our prior and first guess TCWV is 40 mm with a standard deviation of 7.5 mm, although observationally a heuristic band ratio is commonly used to provide a first guess and a locally appropriate prior would be selected. This choice of prior does not change our derived spatial statistics (Supplement Fig. 3), although it results in a small shift of mean retrieved TCWV and reflectance (e.g. posterior TCWV shifts by 0.15 mm when the prior is shifted by 32.5 mm).

### 3.1.2 Profile subsets, emulator development and sensitivity tests

All retrievals use radiances simulated at SZA $= 45°$, using the profiles associated with an individual footprint and assuming a plane-parallel atmosphere. We define "clear sky" as where cloud water path $< 1 \times 10^{-3}$ mm, approximately $\tau < 0.3$ in a typical sub-adiabatic cloud (e.g. Szczodrak et al., 2001). Clear-sky columns are ranked by TCWV, and 101 columns equally spaced in terms of this ranking are taken (Supplement Fig. 4 justifies $N = 101$).

All snapshots in a given LES case are combined and Isofit-retrieved TCWV$_{ret}$ is used to fit an emulator in combination with the forward-model TCWV via

$$\text{TCWV}_{ret} = a_1 \text{TCWV} + a_2 + \epsilon, \tag{2}$$

where $a_1$ and $a_2$ are the trend and intercept parameters from an optimised-least-squares fit and $\epsilon$ is random zero-centred Gaussian noise with standard deviation from the emulator fit residuals. Tests with SZA from 14 to 60° show no significant differences in $a_1$ with SZA, while the standard deviation of $\epsilon$ increases by up to 25 % at SZA $= 60°$ relative to SZA $= 45°$ (Supplement Fig. 5, Supplement Table 1). Section 3.1.4 shows how we are able to identify and remove the effect of $\epsilon$ on derived statistics, so given that $a_1$ did not change with SZA, we anticipate that our conclusions will largely apply to SZA up to and including 60°.

Forward-simulation AOD varied from 0.1 to 0.2, and most footprints were assigned AOD $= 0.2$. Supplement Figs. 6–7 show weak sensitivity of retrieved TCWV to AOD. The analysis is separated into two parts: Sect. 3.2.1 shows results for sensitivity of TCWV$_{ret}$ to changes in the surface spectrum within selected ARM snapshots and Sect. 3.2.2 shows changes in retrieved TCWV over a single surface type for all snapshots.

### 3.1.3 Development and fitting of the retrieval emulator

For each emulator we use all snapshots within an LES run to fit Eq. (2) (separate snapshot fits do not affect conclusions, Supplement Fig. 8), and full-snapshot fields of TCWV$_{ret}$ are then emulated using Eq. (2) with LES TCWV as input. The surface analysis uses the first three ARM snapshots and seven surface spectra from the Isofit surface model clusters, three of which are typical of vegetation and the others of mineral surfaces. The database used to generate the surface model includes artificial surfaces, which are largely captured by the "mineral" spectra. An additional test was run with ARM_18000s profiles over uniform Lambertian

surfaces with $\rho_s = 0.1$–$0.5$ in increments of 0.1. The atmospheric analysis uses the MODTRAN cropland and ocean $\rho_s$ spectra for all 23 snapshots, although poor performance over dark surfaces means that the main emulator results are reported only for the land-surface retrievals.

Figure 3a shows typical spectra simulated over several surfaces: notably, the MODTRAN $\rho_s$ spectra have sharp changes that are not included in the Isofit surface model and therefore provide a challenging test of the retrieval code's ability to retrieve TCWV outside of the surface conditions for which it was developed.

With regards to the emulator parameters, non-unity $a_1$ represents biases in the local retrieval sensitivity $\mathrm{dTCWV_{ret}/dTCWV}$. Possible causes will be discussed in Sect. 3.2.3, but this is the main concern for retrieval of local variability statistics because the retrieved standard deviation will be scaled by $a_1$, and this scaling will be undetectable in the absence of independent validation data. Changes in $a_1$ also change the derived spatial $r^2$, since $a_1 > 1$ increases retrieved $\sigma_x$ variance and will increase $r^2$. The parameter $a_2$ is related to a combination of the mean bias and the magnitude of $a_1$ within a snapshot and may depend on factors such as surface type or biases in the LUT-assumed $T$ and $q$ profiles as seen for MERIS retrievals in Lindstrot et al. (2012). For our spatial statistics, $a_2$ has no effect since it is subtracted during calculation. The parameter $\varepsilon$ represents non-systematic errors within a scene.

Importantly, $\sigma_\varepsilon$ is not the typical error seen in validation or inter-comparison exercises (Diedrich et al., 2015; Nelson et al., 2016; Pérez-Ramírez et al., 2014), since in these studies the varying biases between products in different conditions will add to the reported errors and make them larger than the $\sigma_\varepsilon$ appropriate for our retrievals.

### 3.1.4 Estimating random error from retrieved fields

Random retrieval error $\varepsilon$ with standard deviation $\sigma_\varepsilon$ adds variance and therefore reduces $r^2$ while adding a high-bias term to estimated $\sigma_x$. Knowing $\sigma_\varepsilon$ would allow removal of its bias contribution to $\sigma_x$, and clearly interpretation of spatial variability at a footprint level requires that $\sigma_\varepsilon$ is small relative to $\sigma_x$. TCWV variability between columns separated by 50 m in the horizontal is far smaller than at larger separations. We will exploit this to estimate the spatially constant $\sigma_\varepsilon$ using an approach based around the second-order structure function $S_2$. Here we describe the recipe and mathematical justification; see Supplement Figs. 9–10 for a step-by-step illustration. For a TCWV field,

$$S_2(\Delta r) = E[(\mathrm{TCWV}(x + \Delta r) - \mathrm{TCWV}(x))^2]. \tag{3}$$

This is the variance between pairwise footprints separated by the distance $\Delta r$, and retrieved $S_2$ includes contributions from the spatial variance characteristic at that separation, $\sigma_x^2(\Delta r)$, and the observational uncertainty $\sigma_\epsilon^2$. The subtraction removes the field mean $\overline{\mathrm{TCWV}}$, and each of the terms

$\mathrm{TCWV}(x)$ and $\mathrm{TCWV}(x + \Delta r)$ will contribute $\sigma_x^2(\Delta r) + \sigma_\epsilon^2$ to the variance. We treat these as independent, so their variances add to give the retrieved $S_{2,\mathrm{ret}}$:

$$S_{2,\mathrm{ret}}(\Delta r) = 2\sigma_x^2(\Delta r) + 2\sigma_\epsilon^2. \tag{4}$$

For ARM_18000s, $\sigma_x(\Delta r = 50\,\mathrm{m})$ is 0.03 mm, compared with the full-snapshot $\sigma_x$ of 0.29 mm. We exploit the smallness of $\sigma_x$ at small $\Delta r$ by smoothing the field in one direction with no overlap between smoothed footprints and then calculating the structure function at $\Delta r = 1$ footprint (20–50 m, depending on the LES) perpendicular to the smoothing direction. For $n$-footprint smoothing, the independent component of variance shrinks by $1/n$, which we attribute to $\sigma_\epsilon^2$. The steps are the following.

(i) Select a direction and evaluate $S_2(\Delta r)$ in that direction for $\Delta r = 1$ footprint separation.

(ii) Smooth the field in the direction perpendicular to $\Delta r$ by averaging over $n_{\mathrm{foot}} = 2$ footprints.

(iii) Recalculate $S_2(\Delta r, n_{\mathrm{foot}} = 2)$, treat the calculated value (i) as $S_2(\Delta r, n_{\mathrm{foot}} = 1)$, regress $S_2(\Delta r, n_{\mathrm{foot}})$ against $1/n_{\mathrm{foot}}$, and take the best-fit trend to be equivalent to $2\sigma_\epsilon^2$.

By smoothing in one direction and then calculating orthogonally, the separation distance $\Delta r$ does not grow with smoothing, and so we maintain the advantages of the small $\sigma_x(\Delta r = 20$–$50\,\mathrm{m})$. To estimate TCWV $\sigma_\varepsilon$ with EMIT-like $\Delta x$, this method outperforms a standard spatial smoothing filter approach (Supplement Fig. 11).

### 3.1.5 Calculating spatial statistics and relationship with spatial smoothing

We calculate the spatial standard deviation $\sigma_x$ of clear-sky TCWV and $\mathrm{TCWV_{ret}}$ for each snapshot. The random error $\sigma_\varepsilon$ is then estimated following Sect. 3.2.3 and subtracted in quadrature:

$$\sigma_{x,\mathrm{ret,corr}} = \sqrt{\sigma_{x,\mathrm{ret}}^2 - \sigma_{\epsilon,\mathrm{ret}}^2}, \tag{5}$$

where the subscript "ret" means retrieved and "corr" means corrected.

The other target statistic is $r^2$ between TCWV and $\mathrm{TCWV_{ret}}$; we calculate this directly and also estimate it via

$$r_{\mathrm{est}}^2 = \frac{\sigma_{x,\mathrm{ret}}^2 - \sigma_{\epsilon,\mathrm{ret}}^2}{\sigma_{x,\mathrm{ret}}^2}. \tag{6}$$

Where emulator trend $a_1 = 1$, estimated error from Sect. 3.2.3 is accurate, and there are no spatially varying errors, Eq. (6) should reproduce retrieval $r^2$. However, $a_1 \neq 1$ means each $\sigma_{x,\mathrm{ret}}^2$ term will be multiplied by $a_1^2$, resulting in an erroneous $r^2$ estimate. User requirements for $r^2$ will depend on application; we arbitrarily select $r^2 = 0.9$ as a target

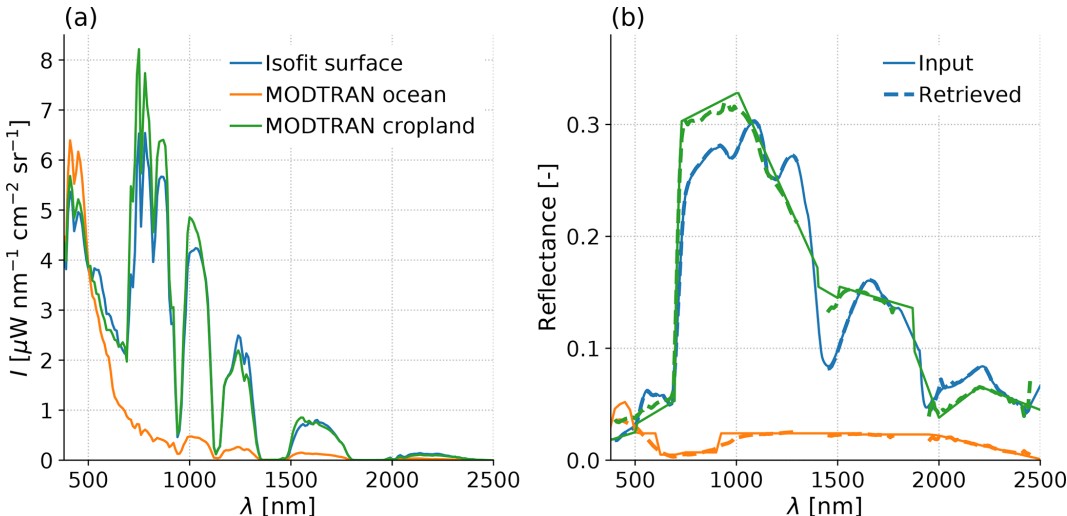

**Figure 3.** Examples of **(a)** simulated spectra and **(b)** used surface reflectances in the forward model (solid lines) and those retrieved by Isofit using EMIT instrument characteristics (dashed lines). Each colour refers to a surface type as listed in the panel **(a)** legend.

and compare Eq. (6) estimates with the true field values. True $r^2$ is unknowable without perfect knowledge of the TCWV field, but operational estimation using Eq. (6) would allow users to determine whether their requirements are likely to be met.

If $r^2$ is too low for the desired application, then averaging over footprints may address this; although it results in loss of fine spatial information, it may be necessary to suppress errors or may be enforced by effective horizontal smearing where SZA $> 0°$.

We show the results of sequentially smoothing the TCWV and TCWV$_{\text{ret}}$ field on both $\sigma_x$ and $r^2$ and smooth in both horizontal directions, for example going from $50\,\text{m} \times 50\,\text{m}$ to $100\,\text{m} \times 100\,\text{m}$. Smoothed footprints do not overlap and so are independent, and the smoothing is done on TCWV$_{\text{ret}}$ rather than on the radiance field. This avoids the requirement for additional forward-model runs and furthermore allows predictions of how $r^2$ changes with resolution by applying Eq. (6) with a minor modification:

$$r^2_{\text{est}} = \frac{\sigma^2_{x,\text{ret}} - \frac{\sigma^2_{\epsilon,\text{ret}}}{n}}{\sigma^2_{x,\text{ret}}}, \tag{7}$$

where $n$ is the number of footprints over which TCWV$_{\text{ret}}$ has been smoothed, e.g. for the $50\,\text{m} \times 50\,\text{m}$ to $100\,\text{m} \times 100\,\text{m}$ transition $n = 4$. In this case, $\sigma_{\varepsilon,\text{ret}}$ must be calculated at the native resolution and therefore exploits the smaller TCWV variance at $\Delta r \sim 50\,\text{m}$ rather than the higher variance in a smoothed field with larger $\Delta r$.

## 3.2 Simulated retrieval results

### 3.2.1 TCWV retrievals over different surfaces

We first remind readers that "retrieval error" here only includes errors present in these synthetic retrievals and excludes several real-world sources, such as how the true atmosphere is not plane-parallel as assumed in our radiative transfer. Retrieved surface $\rho_s$ spectra and TCWV$_{\text{ret}}$ versus forward-model TCWV are shown in Fig. 4. Surface $\rho_s$ are retrieved well, with mean bias magnitude equivalent to 0.2 %– 1.6 % of true $\rho_s$ (e.g. for Lambertian $\rho_s = 0.1$, the mean bias is 0.00021), and standard deviation across all channels is 2 %–4 % of true $\rho_s$. The largest contribution to errors is from spikes near $\lambda \sim 2.06\,\mu\text{m}$. Inspection found that the MOD-TRAN $CO_2$ concentration changes between default profiles versus prescribed $T$ and $q$ profiles. In future an up-to-date $CO_2$ mixing ratio will be assigned, but the higher LUT value (361 ppmv) versus the forward-model value (323 ppmv) results in the retrieval overly brightening the surface in the strong $CO_2$ band near $\lambda \sim 2.06\,\mu\text{m}$.

Comparing Fig. 4d–f, TCWV$_{\text{ret}}$ over mineral surfaces is a mean 0.44 mm higher than over vegetation. From panel f, some of this difference is likely related to mean surface brightness: darker surfaces give higher TCWV$_{\text{ret}}$. The other differences in TCWV$_{\text{ret}}$ between surfaces must be due to spectral shape, but it appears that surface-induced errors are small when considering only mixed vegetation or mixed mineral surfaces. Regardless of the surface, a bias of order $\sim 1\,\text{mm}$ remains, which is similar to the largest difference introduced by surface type and may be related to other retrieval errors such as inappropriate atmospheric profile shapes assumed in the LUT. However, the derived spatial statistics we are interested in here are not affected by any mean bias.

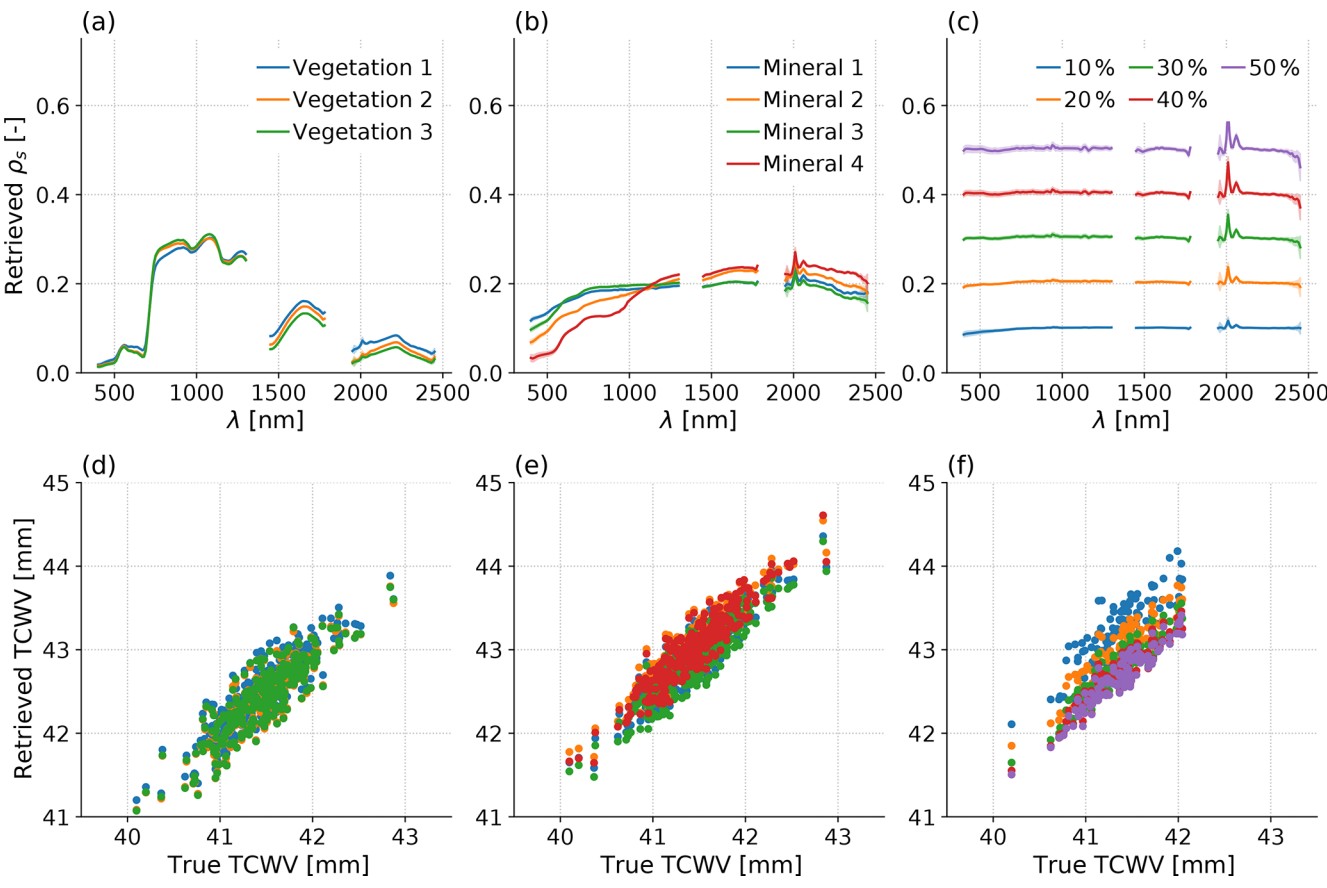

**Figure 4. (a)–(c)** Retrieved reflectance spectra for **(a)** vegetation, **(b)** mineral and **(c)** spectrally uniform surfaces. Lines show the mean of all simulated retrievals and shading shows $\pm 1\sigma$. **(d)–(f)** Retrieved TCWV as a function of true TCWV for the same. The vegetation and mineral cases use three snapshots ($N = 303$) and the Lambertian surfaces just use ARM_18000s ($N = 101$).

Figure 5 shows example scenes with different surface types. The true TCWV standard deviation $\sigma_x$ is 0.28 mm (panel a), while over the uniform surfaces the retrieval gives 0.33 mm (panels b and c), with the larger value due mainly to the $\sigma_\varepsilon$ contribution. Over the striped surfaces it is 0.40 mm (panel d) due to the additional variance from combining surface types. However, if the top or bottom half of panel d is selected, then both return $\sigma_x$ of 0.33 mm, i.e. the same as over a fixed vegetation or mineral surface. Statistics should not be taken over scenes with both vegetation and mineral surfaces, but the Isofit surface classification, which is output by the retrieval, should be used to identify areas of sufficiently similar surface type for calculation of TCWV spatial statistics. The rest of the analysis assumes the MODTRAN cropland default surface.

### 3.2.2 TCWV retrievals over vegetation surfaces in all LES snapshots

Figure 6 shows TCWV retrievals over the MODTRAN cropland and ocean surfaces. The poor performance over ocean absent sun glint justifies our land-only investiga-

tion. Over land the mean bias ranges from $-3.0\%$ (DRY) to $+1.8\%$ (BOMEX), while the within-scene $\sigma_\varepsilon$ is from 0.52 % (ARM_lsconv) to 0.67 % (BOMEX). As discussed in Sect. 3.1.3, VSWIR TCWV validation studies typically report error metrics larger than our $\sigma_\varepsilon$, but their values include inter-product differences in bias, which are potentially far larger. Bias is indeed sensitive to the assumed meteorological profiles, since re-running the ARM_18000s retrievals using a LUT developed with the MODTRAN default "tropical" atmospheric profile shifts the mean bias from $0.33 \pm 0.04$ mm to $0.14 \pm 0.04$ mm (mean $\pm 2\sigma$).

For the purpose of spatial variability in TCWV at scales of tens of kilometres, the distinction between large-area and small-area retrieval errors is important. Generally speaking, the error in an individual column TCWV retrieval is of order 2 %–3 % since that includes the bias term, but for estimates of sub-10 km spatial variability, the within-LES TS5 0.5 %–0.7 % is the error of interest.

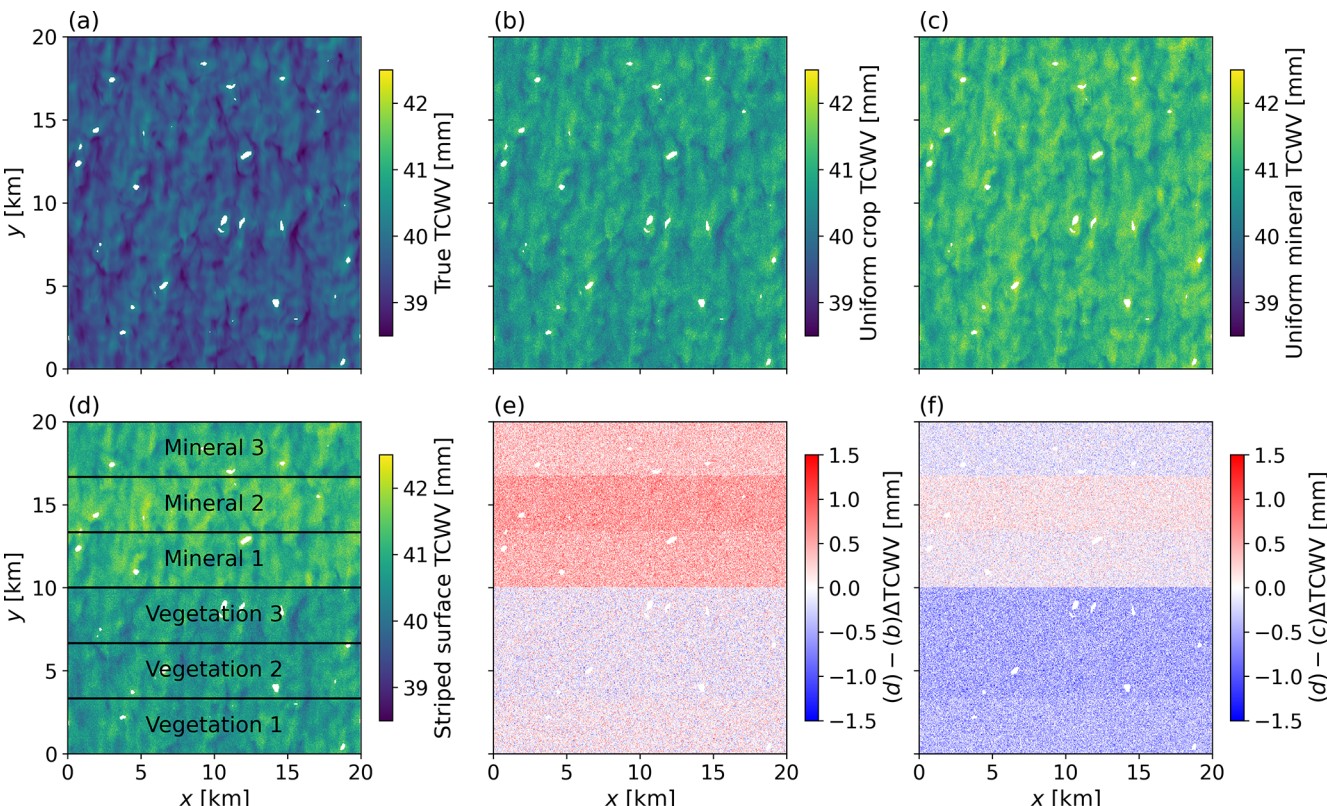

**Figure 5.** ARM_18000s **(a)** true TCWV, **(b)** retrieved TCWV over a uniform vegetated surface, **(c)** retrieved TCWV over a uniform mineral surface, **(d)** retrieved TCWV over stripes of uniform surface types as labelled in the figure, **(e)** difference induced in retrieved TCWV by surface type relative to mixed vegetation as **(d)** minus **(b)**. **(f)** Difference relative to a mixed mineral surface as **(d)** minus **(c)**. Clouds are masked in all cases. Panel **(f)** represents **(d)** minus **(c)**.

### 3.2.3 Emulator parameters

Emulator parameters with $\pm 2\sigma$ confidence intervals are listed in Table 2, and significant ($p < 0.05$) non-unity trends can be seen most clearly for BOMEX (green) and RICO (purple) in Fig. 6a; the retrieved properties are more variable than reality, with trends of 1.34 and 1.22 mm mm$^{-1}$, respectively. Meanwhile, the ARM and ARM_lsconv trends are both $< 1$ mm mm$^{-1}$. Therefore $\sigma_x$ calculated for BOMEX will be 34 % too high and for ARM 6 % too low.

We argue that the most likely causes of emulated trend bias are related to the vertical $T$ and $q$ profile. Firstly, d$I$/d$q$ is non-linear and varies with atmospheric conditions due to line broadening and interaction with aerosol layers. The $a_1$ fit parameter may therefore be sensitive to differences between true profiles and those assumed in the retrieval LUT. Secondly, the LUT uniformly scales $q(z)$ profiles, whereas the horizontal variability in $q$ tends to peak at specific altitudes (Fig. 1).

Two tests provide some evidence for this. Firstly, when using different standard atmospheres to generate lookup tables for the DRY case, the retrieval gradient changes by 5 %, and secondly, when re-running all BOMEX retrievals with

**Table 2.** Emulator parameters relating true TCWV to TCWV$_{ret}$. In Eq. (2) the trend is $a_1$, the intercept is $a_2$, and residual $\sigma$ is the standard deviation used in generating the samples of $\epsilon$. Values are shown $\pm 2\sigma$.

| Case | Trend (mm mm$^{-1}$) | Intercept (mm) | Residual $\sigma$ (mm) |
|---|---|---|---|
| ARM | $0.94 \pm 0.02$ | $0.29 \pm 0.07$ | $0.22 \pm 0.03$ |
| ARM_lsconv | $0.97 \pm 0.04$ | $0.14 \pm 0.17$ | $0.23 \pm 0.03$ |
| BOMEX | $1.34 \pm 0.06$ | $-1.15 \pm 0.22$ | $0.20 \pm 0.03$ |
| DRY | $1.13 \pm 0.03$ | $-0.33 \pm 0.07$ | $0.10 \pm 0.01$ |
| RICO | $1.22 \pm 0.04$ | $-0.77 \pm 0.15$ | $0.21 \pm 0.03$ |

forward radiances generated using the same $q$ profile shape that has been scaled to match the original range of TCWV, the retrieval gradient changes by 9 % (Supplement Fig. 12). These results suggest that retrievals could be improved by using more accurate meteorological profiles in the LUT development and by using a more appropriate scaling for $q$ as a function of $z$ in the LUT.

### 3.2.4 Snapshot statistics and estimation of random error

Figure 7a shows how $\sigma_x$ of TCWV$_{\mathrm{ret}}$ is overestimated in every snapshot (circles). Figure 7b shows that the estimated retrieval error $\sigma_\varepsilon$ agrees excellently with the truth, and after removing $\sigma_\varepsilon$, the triangles in Fig. 7a show the consistent overestimate is removed. Random error, such as that introduced by some instrumental uncertainties, can be precisely identified and removed from the spatial variance calculations. To estimate $\sigma_x$, the largest error source we consider is due to emulator slope. Other potential sources would be due to surface variation, which can be mitigated by selecting regions of similar surface classification as suggested in Sect. 4.1, and due to spatially varying errors, such as inter-pixel calibration biases or those induced by unmodelled temperature gradients across the sensors. Separate approaches are required to account for these issues.

Next, we consider the $r^2$ coefficient between TCWV and TCWV$_{\mathrm{ret}}$, with an illustration in Fig. 8, where the RICO_14400s TCWV$_{\mathrm{ret}}$ fields are shown at the native resolution and after smoothing down to 80 m. The random retrieval error is visible as speckling (Fig. 8a and b) and clearly reduces following smoothing. The 2D histograms (Fig. 8c and d) demonstrate the increase in $r^2$ from 0.82 to 0.95 following a coarsening of the $\Delta x$ resolution from 40 to 80 m.

Figure 9 summarises the true and estimated statistical values as horizontal resolution is sequentially degraded. Comparison of Fig. 9a and b reveals that there is only a small decrease in $\sigma_x$ as resolution coarsens up to hundreds of metres, and the biases between estimated and true values follow emulator $a_1$ trends as expected, with ARM and ARM_lsconv too low and DRY, RICO and BOMEX too high.

Regarding $r^2$ in Fig. 9c and d, Eq. (7) reliably predicts true $r^2$, so a user could determine the spatial resolution required to achieve a desired $r^2$. In all snapshots $r^2 > 0.90$ at 150 m resolution, and in 21 of 23 cases this is achieved at 100 m. Therefore, with the errors accounted for here, the EMIT instrument could capture 90 % of spatial variability in PCWV$_{\mathrm{PBL}}$ at 100 m resolution in the PBL conditions examined here, a factor of 7.5 improvement in the MERIS full-resolution retrievals. However, this conclusion does not account for the spatial smearing caused by SZA.

### 3.3 Discussion of retrieval results and limitations

This section has addressed questions (2) and (3) from Sect. 1 and shown that random errors introduced by EMIT's instrumental error can be accurately identified and removed. Provided that an observed domain consists of mixed vegetation or mixed mineral surfaces, then our derived error in $\sigma_x$ using EMIT is from $-7$ % to $+34$ %. Isofit returns surface type, meaning that such domains can be identified from retrievals.

Computational limitations forced adoption of an emulator approach, which provides a useful framework to assess error sources. Firstly, this framework shows that the errors of interest for retrieval of spatial statistics of PCWV$_{\mathrm{PBL}}$ are the gradient $a_1$, equivalent to dTCWV$_{\mathrm{ret}}$/dTCWV, and random error $\sigma_\varepsilon$. We show that $\sigma_\varepsilon$ can be estimated and removed and that the main error is that in $a_1$, most likely driven by the retrieval's atmospheric profile assumptions, which can be addressed in future development. Our method to derive $\sigma_\varepsilon$ also allows users to predict spatial correlation; in particular, we found that an $r^2 > 0.9$ requirement requires smoothing to 100–150 m resolution. This is a factor of 3–8 improvement in sampling relative to MERIS full resolution.

Limitations include the use of the same radiative transfer code for forward and inverse simulations, so spectroscopic errors were ignored, as were errors in cloud and shadow masking, those caused by topography, or errors that correlate between footprints.

Spectroscopy errors can be estimated (Thompson et al., 2020) and should shrink in future with developments, with ongoing research in water vapour absorption spectroscopy (Elsey et al., 2020; Lechevallier et al., 2018; Menang et al., 2021) and a history of targeted development of spectroscopy to improve retrievals, such as for OCO-2 (Drouin et al., 2016; O'Dell et al., 2018; Payne et al., 2020). The surface remote-sensing community has tools for addressing topography (Kobayashi and Sanga-Ngoie, 2008; Teillet et al., 1982), and there are also approaches to dealing with nearby clouds to minimise the effect of imperfect cloud edge identification, shadowing and 3D cloud-radiative effects (Massie et al., 2021). Nevertheless, these are all topics that are worth evaluating for Isofit-like TCWV retrievals.

We note that our $\sigma_\varepsilon$ is smaller than the errors reported in product intercomparison studies, but those studies implicitly capture variance due to differing mean biases, i.e. the $a_2$ term in our emulator, which is larger than the other terms. An evaluation of our retrieved $\sigma_x$ would require independent validated sources such as passive microwave or differential absorption lidar data with $\Delta x \leq 50$ m that are collocated with VSWIR TCWV$_{\mathrm{ret}}$. Reported comparisons are typically of bias and root-mean-squared error (RMSE) of satellite VSWIR retrievals relative to surface-based or other satellite products and are calculated from datasets across a range of times and sometimes places. Furthermore, the comparison data generally have larger $\Delta x$ and may not be perfectly collocated in time and space, introducing additional variance that contributes to reported RMSE. Typical published analyses include within their RMSE uncertainties due to differences in space and time of measurements and any differences between the $a_2$ terms between the VSWIR and validation dataset retrievals. Therefore, these reported errors cannot be compared with our values, which are calculated within individual LES runs. We can, however, report that our errors are similar to Thompson et al. (2021)'s airborne Isofit retrieval statistics against nearby AERONET surface stations, which reported an RMSE of 2.8 mm. Flight C data from Fig. 9 of that paper show a spatial standard deviation of 0.19 mm when smoothed

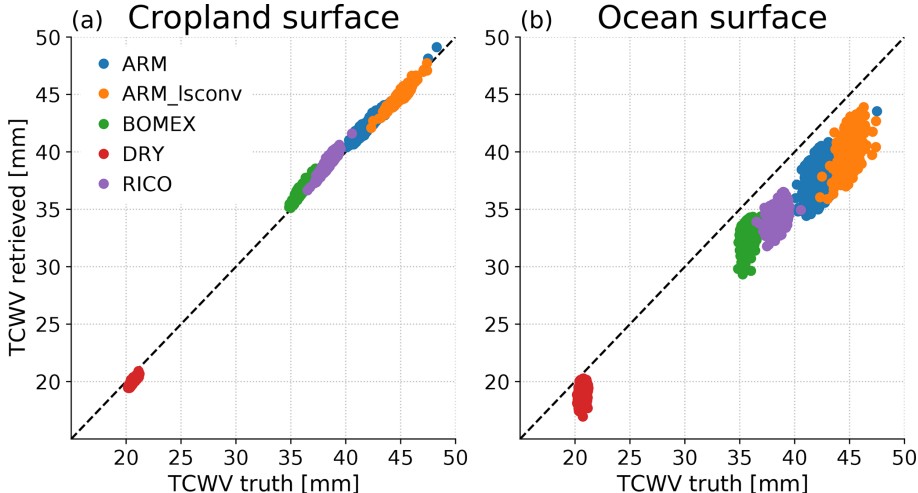

**Figure 6.** Retrieved TCWV as a function of the truth for all snapshots in each LES case over **(a)** cropland and **(b)** ocean. Note that the TCWV values differ from those derived from the LES due to differences in the MODTRAN layer interpolation and calculations.

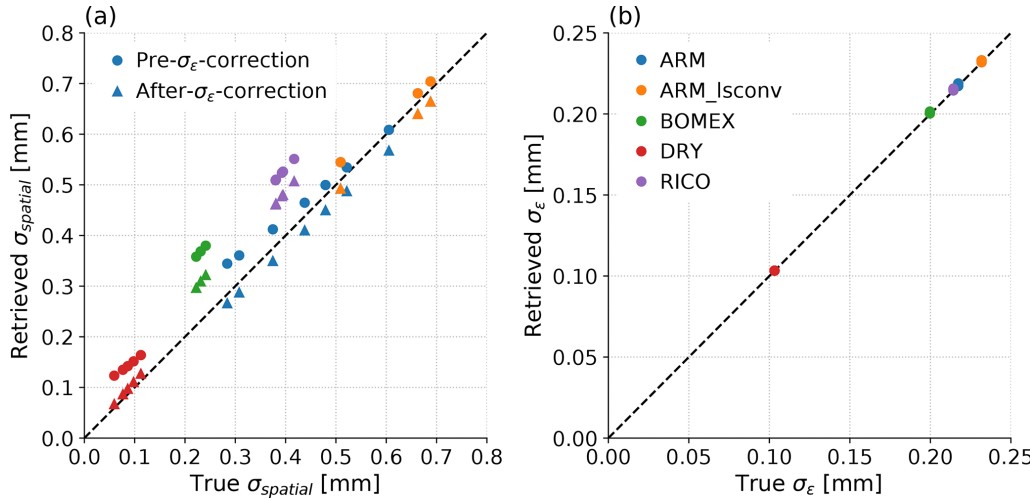

**Figure 7. (a)** Estimated clear-sky horizontal standard deviation as a function of the true value for each snapshot for raw retrievals (circles) and retrievals after removal of the random component of retrieval error, e.g. that induced by instrumental noise (triangles). **(b)** The estimate of retrieval error as in Sect. 3.1.4 as a function of the true error in each case.

to $\Delta x = 48$ m, which is within the LES-simulated range, and $\sigma_\varepsilon$ is estimated at 0.18 mm, although that is not comparable to our values since it uses AVIRIS-NG rather than EMIT and is over ocean sun glint rather than land.

Reported RMSEs over land for other VSWIR instruments include 0.9–1.3 mm for OCO-2 (Nelson et al., 2016), 1.4–3.7 mm for MERIS (Lindstrot et al., 2012), 0.9–2.0 mm for MODIS (Diedrich et al., 2015), 1.3–3.3 mm for OLCI (Preusker et al., 2021) and up to 2.4 mm for Sentinel-2 (Obregón et al., 2019). The range of $\mathrm{TCWV_{ret}}$ simulated in Fig. 4 is therefore consistent with typical errors reported for other instruments. Interestingly, Obregón et al. (2019) report a gradient of 0.9 between Sentinel-2 and AERONET $\mathrm{TCWV_{ret}}$. This is derived from data across multiple sites and times and so cannot be compared to our gradients derived from individual LES cases but indicates that different retrievals may indeed have relationships between TCWV and $\mathrm{TCWV_{ret}}$ which are not 1 : 1, and thus our non-unity $a_1$ parameters, which scale derived $\sigma_x$, are credible.

## 4   Effect of SZA variation on retrieved properties

### 4.1   Calculation of $\mathrm{TCWV_{ret}}$ accounting for the light path at different solar zenith angles

Along-path-integrated water vapour (IWV) for SZA ranging from 0 to 60° inclusive in increments of 15° is calculated using ray tracing. The sunlight's horizontal component is in

## RICO_14400s

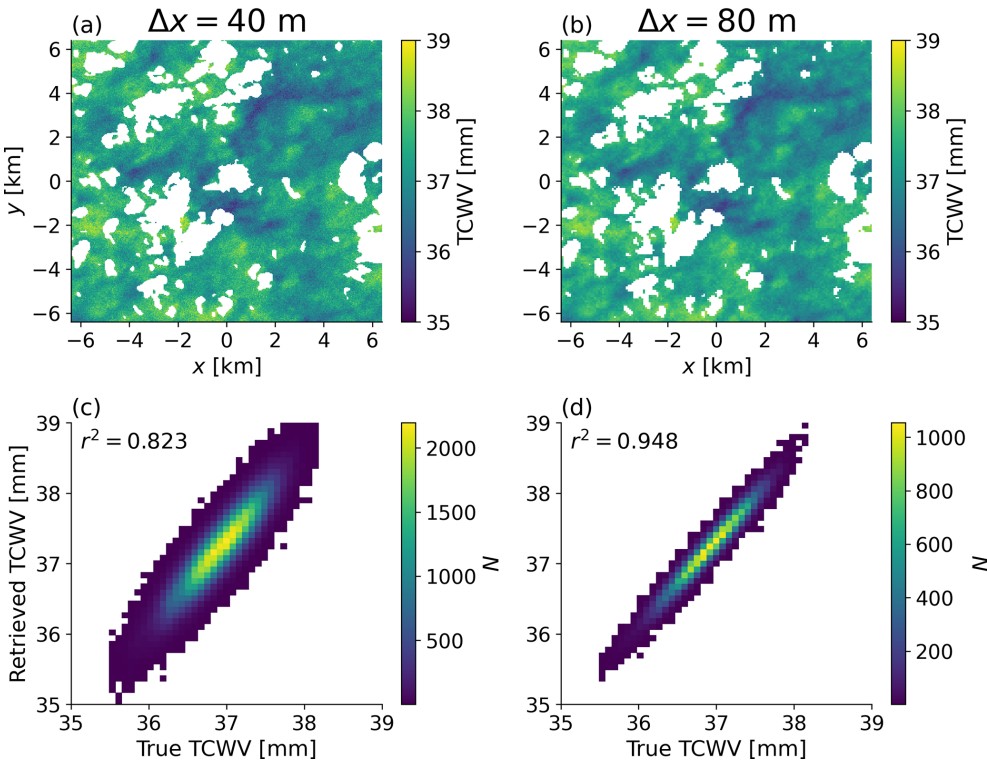

**Figure 8. (a)** Retrieved TCWV at 40 m resolution, **(b)** retrieved TCWV at 80 m resolution, **(c)** 2D histogram of retrieved TCWV as a function of the truth at 40 m resolution, and **(d)** 2D histogram of the same at 80 m resolution. The squared Pearson correlation coefficient, $r^2$, is written in the upper left corner of **(c, d)**.

the negative $y$ direction, viewing zenith angle is $0°$, and the ray is traced from the top of atmosphere to the centre of each surface footprint. Each partial grid cell encountered has its $q$ weighted by the pressure-corrected path through that cell before obtaining IWV. The cloud mask is extended by a "shadow mask" where cloud LWP $> 1 \times 10^{-3}$ mm along the solar direct ray path. This IWV is referred to as a TCWV for consistency with standard retrieval terminology, even though it is not directly a measure of the column over the footprint. The Sect. 3 analysis is then repeated using the same emulators developed using radiative transfer with SZA $= 45°$, a plane-parallel assumption and footprint column profiles. Different SZAs may change the sensitivities somewhat, but we do not expect results substantially outside the range of those considered here.

### 4.2 Effect of SZA variation on retrieved properties

Figure 10a and d show apparent TCWV in ARM_lsconv_36000s (i.e. when convection is most developed) at SZA $= 15$ and $60°$, and in Fig. 10c and d the clear vertical pattern of positive followed by negative biases relative to true TCWV is clear, with greater magnitude and

larger regions of continuous positive or negative bias at higher SZA.

Figure 11 shows that this spatial smearing destroys the correspondence between footprint and path TCWV, with $r^2$ around 0.1 with SZA as small as $30°$. This can be compensated only somewhat by spatial smoothing, as Fig. 12 shows that even footprints degraded to 300 m are affected by SZA. The calculated $\sigma_x$ at SZA $= 0°$ match those from Fig. 9, with biases from the emulator slope parameter in Table 2. Larger SZA in these cases increases the magnitude of this bias, but the difference in $\sigma_x$ as SZA changes from 15 to $60°$ is smaller than the RICO or BOMEX emulator-trend-induced biases. The retrieved $\sigma_x$ with footprint size tracks reality, suggesting that the horizontal distribution statistics might still be captured even at large SZA. Furthermore, the statistical error estimation from Sect. 3.1.4 has effectively identical performance regardless of SZA (not shown).

However, Fig. 12b shows that a VSWIR-retrieved map TCWV$_\text{ret}$ does not accurately represent the actual spatial variability in TCWV and by extension PCWV$_\text{PBL}$ even for SZA $= 15°$, and this is a fundamental limitation caused by the solar path through the atmosphere. In fact, the TCWV$_\text{ret}$

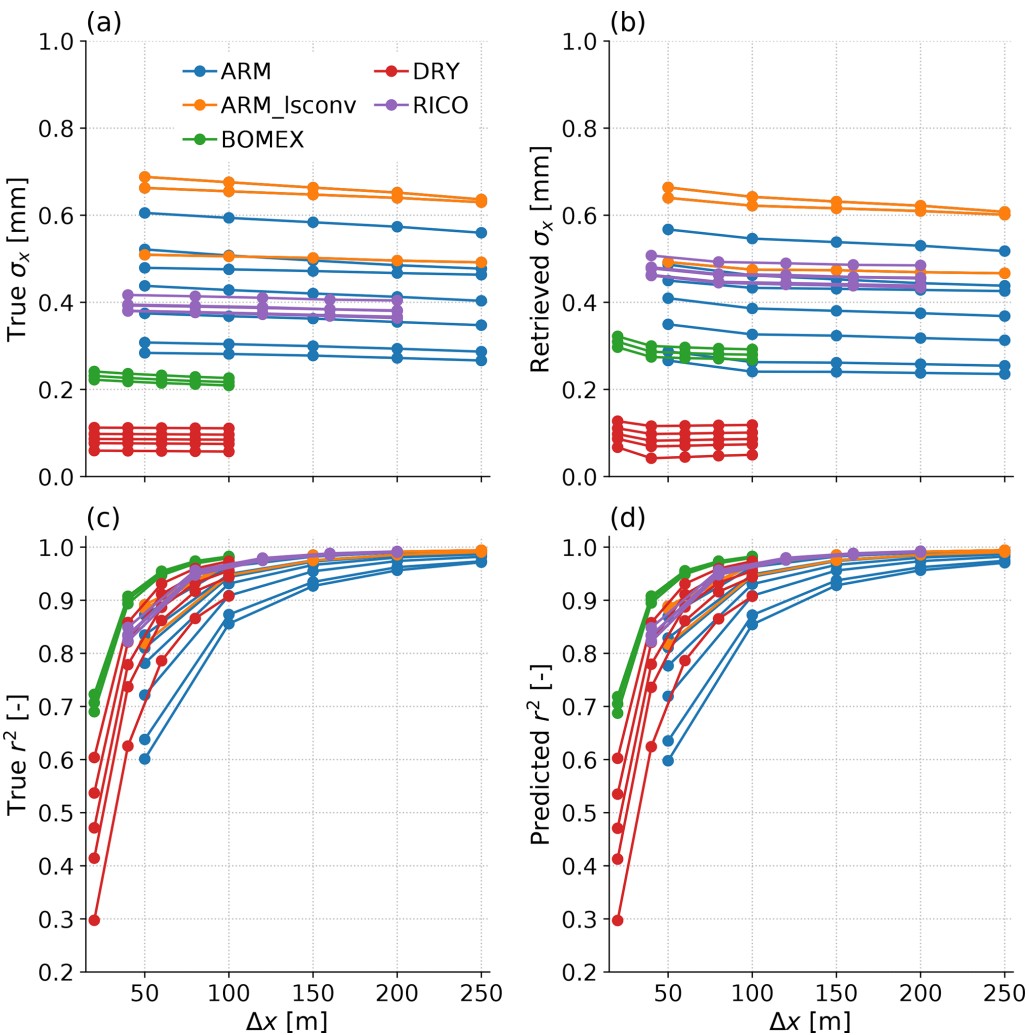

**Figure 9.** Changes in the true and retrieved statistical properties for LES as a function of spatial resolution $\Delta x$. **(a)** The true standard deviation calculated directly from the LES output, **(b)** retrieved standard deviation after removing the estimated retrieval error as in Sect. 3.1.4, **(c)** $r^2$ between true TCWV and $\text{TCWV}_{\text{ret}}$, and **(d)** estimated $r^2$ using Eq. (7).

map corresponds better to the TCWV map at the horizontal location where the downward solar path enters the PBL, but improvement in $r^2$ is limited (Supplement Fig. 13).

Figure 13 shows that while the retrieved TCWV distributions are biased, as previously discussed, SZA increases cause only minor visible changes in distribution shape. This indicates that important statistics of the TCWV (and therefore $\text{PCWV}_{\text{PBL}}$) field can be obtained at the native footprint resolution, despite the poor correspondence of any individual footprint to the column located at that position. The primary advantages of finer spatial resolution are that (i) it allows better calculation of $\sigma_\varepsilon$ than at coarser resolution using Sect. 3.1.4's method, due to the smaller $\Delta r$ between footprints and, (ii) when calculating statistics such as standard deviation on local scales, statistical errors are reduced by the larger number of footprints. For example, $\Delta x = 50$ m represents approximately 25 times more measurements than

MODIS or MERIS. If standard deviation were desired for a $1\,\text{km} \times 1\,\text{km}$ region, $N = 16$ from 250 m footprints results in a sampling error of $\pm 17.7\,\%$ versus $\pm 3.5\,\%$ for $N = 400$ from 50 m footprints.

## 5 Discussion and conclusions

Modern and upcoming VSWIR instruments promise unprecedented horizontal resolution for the study of surface properties, with emphases ranging from mineral regions that are the source of dust (EMIT) to routine observation of agriculture and biodiversity (CHIME) to the full spectrum of study under the NASA 2017 Decadal Survey's Surface Biology and Geology (SBG) designated observable.

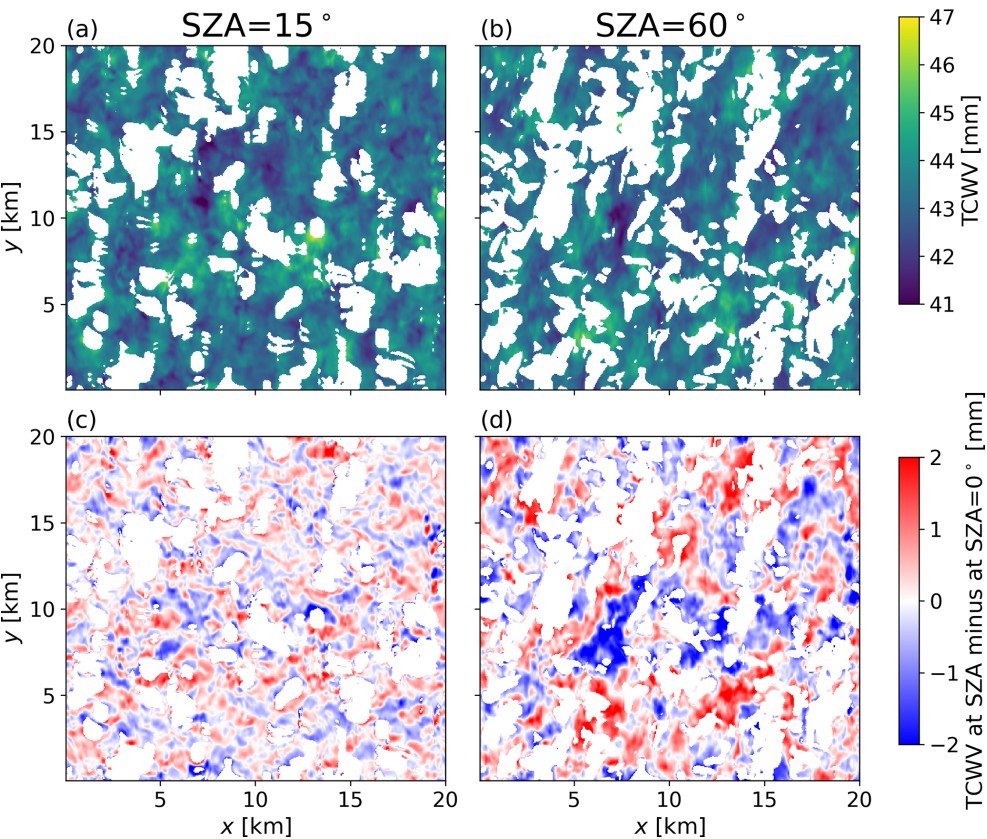

**Figure 10.** ARM_lsconv_36000s-integrated water path calculated along **(a)** SZA = 15° and **(b)** SZA = 60° with the upward path directly up at zenith angle 0°; values labelled TCWV in colour bar for simplicity. Panel **(c)** shows the difference for each footprint by subtracting the true TCWV at SZA = 0° from panel **(a)**, and panel **(d)** shows the same for subtracting the SZA = 0° value from the SZA = 60° value. The "cloud" mask in each case is now extended to include cloud shadows, and the illumination comes from the top of each panel; i.e. sunlight travelling down through the atmosphere has a component in the negative $y$ direction.

This study suggests potential synergies with the Decadal Survey's PBL targeted observable by showing that PCWV$_{PBL}$ variability at high spatial resolution can be inferred using the TCWV$_{ret}$ that will be obtained from EMIT observations. While these measurements lack the vertical resolution that is necessary to advance PBL science, they provide a unique constraint on the mesoscale moisture variability and aggregation within the convective PBL. This analysis is restricted to daytime convective PBLs over land surfaces, which excludes deep convection but still represents a large fraction of meteorological conditions in the tropical to mid latitudes. Importantly, these are the precise conditions in which it is suspected that PBL moisture aggregation influences the timing of deep convective events. Furthermore, given the large number of scenes in which we expect to be able to derive these spatial statistics, these observations could prove useful for constraining the manner in which small-scale variability is parameterised

in shallow convection or unified parameterisation schemes. The Isofit development team has curated additional spectra for a universal prior that includes cryosphere surfaces, but additional work may be necessary to evaluate TCWV over snow, since there is a snow absorption feature near $\lambda = 1\,\mu m$ whose depth depends on snow grain size (Painter et al., 2007) and which overlaps $q_v$ absorption. This may introduce surface–atmosphere covariance that affects the retrieval.

NASA's 2017 Decadal Survey encourages multi-instrument applications, and the VSWIR retrievals discussed here could be combined with radio occultation, thermal infrared (TIR) or passive microwave sounders, which have far larger horizontal resolution but obtain vertical profiles. Early explorations of joint VSWIR-TIR retrievals are promising, suggesting that the sensors provide complementary information on both atmospheric and surface properties. TS6 VSWIR could provide a prior constraint on TCWV in a collocated TIR retrieval, or the TIR-retrieved PCWV$_{upper}$ could be

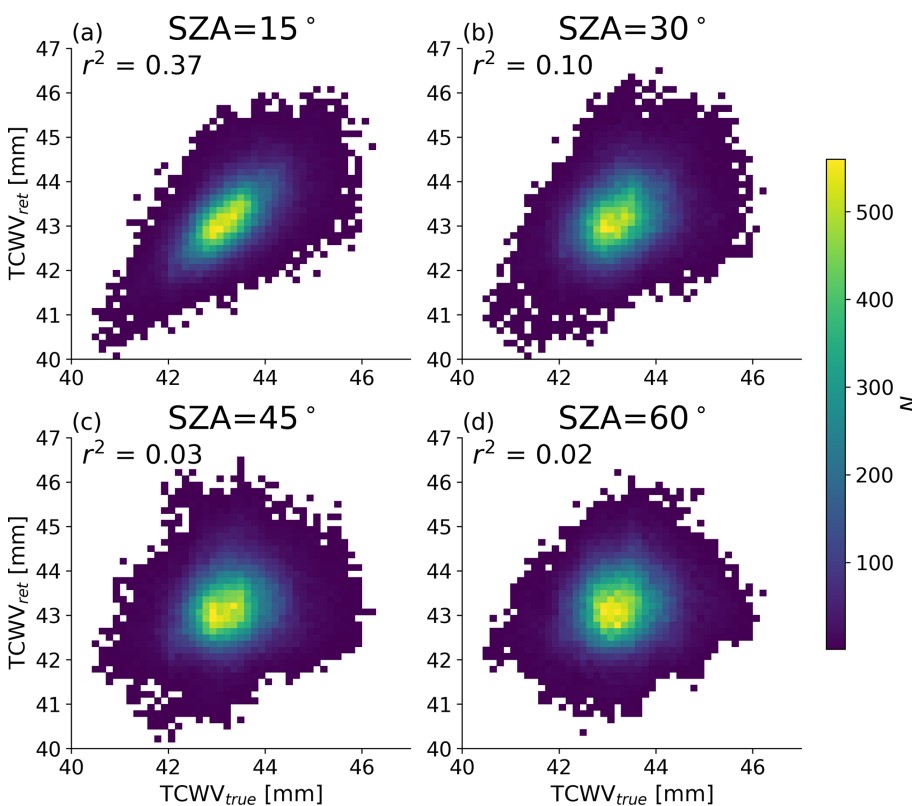

**Figure 11.** 2D histograms between clear-sky TCWV (true value integrated only in column over footprint) and the retrieved values at the corresponding footprint with SZA of **(a)** 15°, **(b)** 30°, **(c)** 45°, and **(d)** 60°. The $r^2$ coefficient is in each panel, and the footprint resolution is the native output of $\Delta x = 50$ m.

subtracted from VSWIR TCWV to estimate $PCWV_{PBL}$, with VSWIR also providing the horizontal statistics of clear-sky $PCWV_{PBL}$ variability within the TIR footprint. Another opportunity is to use coincident TIR-retrieved profiles of $T$ and $q$ to either build a more appropriate LUT for the VSWIR retrieval or to select from among pre-computed LUTs.

In Isofit, the atmospheric component contributes a bias to $dTCWV_{ret}/dTCWV$ and may be the largest source of our errors in $\sigma_x$, which range from $-7\%$ to $+34\%$ of true $\sigma_x$. Development allowing the use of prescribed profiles and the ability to assign variability in $q$ to lower altitudes rather than uniform scaling at all altitudes should reduce these errors, as accounting for temperature reduced biases in MERIS $TCWV_{ret}$ (Lindstrot et al., 2012).

This study also showed how SZA as small as 15° significantly degrades the accuracy of retrieved spatial patterns in TCWV, even at coarser resolutions similar to current sensors such as MERIS. However, the TCWV distribution was far less sensitive to SZA. While our results should strongly affect the interpretation of retrieved maps of TCWV from instruments like MERIS, they suggest that moments of the $PCWV_{PBL}$ distribution can be obtained at unprecedented

horizontal resolution, which may be of use to developers of modern PBL schemes that use or assume such moments. We note that the LES TCWV distributions and their variation with spatial scale may not be realistic, since they tend to be overly dissipative on scales $\leq 6$ grid cells (Bryan et al., 2003), but it is not clear that these biases affect our conclusion regarding the ability to obtain distributional statistics that represent horizontal variability at scales as small as 50 m.

Future work could address uncertainties that are ignored here, such as topography or cloud 3D radiative effects via 3D radiative transfer simulations which avoid several of our assumptions, such as a plane-parallel atmosphere. A particular limitation is that this analysis did not consider vertical structure or PBL height beyond using that derived from the LES mean profiles. In reality there may be errors in locally estimated PBL height, or that obtained from other sensors may be inconsistent with the $\max(d\theta/dz)$ value used here, and targeted research on this topic would be worthwhile. Observational evaluation of these uncertainties could be performed using collocated airborne measurements of column water vapour from VSWIR and other instruments such as differential absorption lidar or passive microwave imagers,

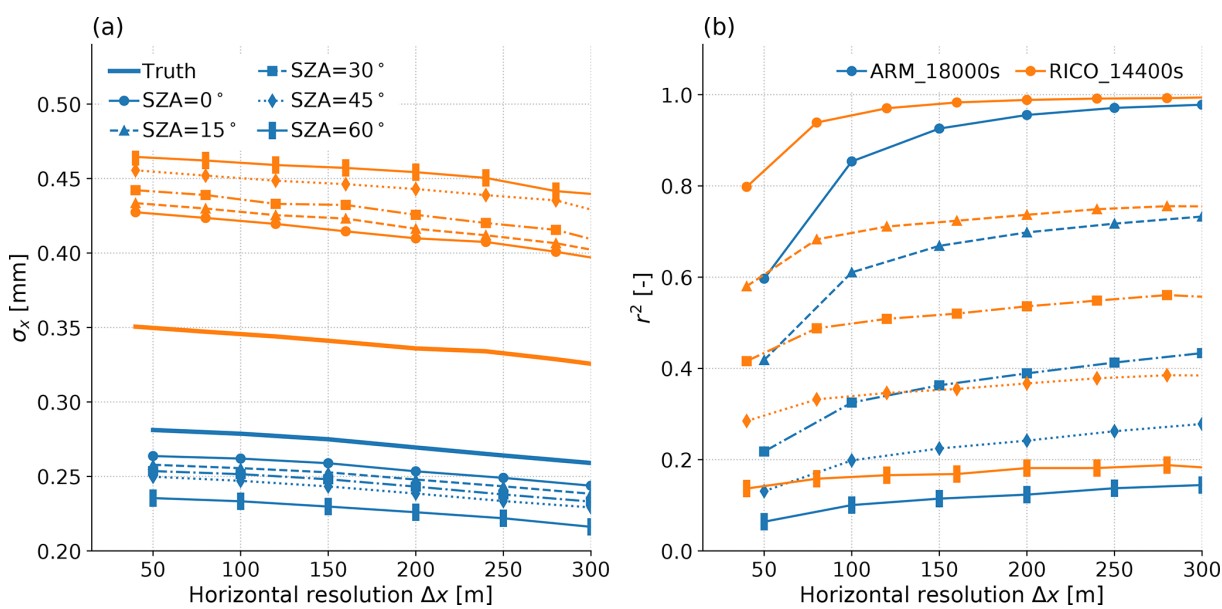

**Figure 12.** Clear-sky TCWV horizontal spatial statistics calculated for ARM_18000s (blue) and RICO_14400s (orange) as a function of the horizontal footprint size. **(a)** Standard deviation $\sigma_x$ as in Fig. 9 and including the random error correction from Sect. 3.1.4. **(b)** Correlation coefficient between column true TCWV and that retrieved for the same footprint as SZA changes. Each line style represents a different SZA as labelled in the legend of **(a)**.

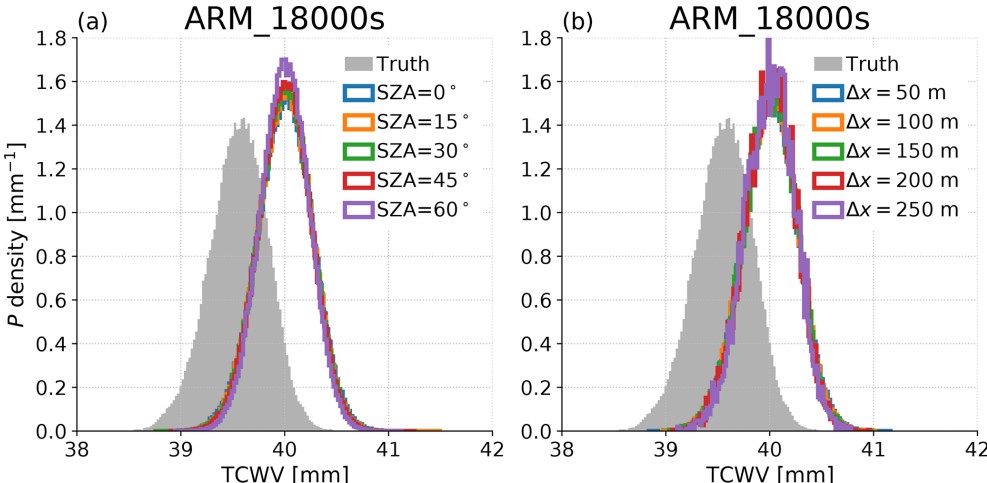

**Figure 13.** Histograms of footprint-estimated clear-sky TCWV, with the truth shown in grey shading. The retrieval estimates are all scaled to remove the variance due to estimated random error. **(a)** Variation with SZA calculated at footprint size $\Delta x = 50$ m and **(b)** variation with footprint size at SZA $= 0°$. In both panels the blue histograms are the same.

provided they can obtain sufficiently high spatial resolution. Finally, this work could be extended to other sensors, such as MSI on Sentinel-2, which is not hyperspectral but provides an exceptionally fine $\Delta x$ of approximately 20 m. Additional high-resolution analysis may be required for this, since Fig. 9a and b imply increases in retrieved $\sigma_x$ at $\Delta x = 20$ m for the two simulations that were run at that resolution.

Despite these caveats, we have shown ways in which atmospheric correction outputs of surface property retrievals for EMIT can provide unique information on fine-scale PBL

water vapour variability and also identified specific development tasks to improve the quality of its atmospheric outputs. With current tools it therefore seems likely that missions such as EMIT and CHIME, which are primarily designated as targeting surface observables, can provide unique information to the atmospheric science community.

*Code availability.* The Isofit retrieval package is available on GitHub (https://github.com/isofit/isofit, last access: 2 June 2021)

(https://doi.org/10.5281/zenodo.4614338, Brodrick et al., 2021 TS7) and MODTRAN from Spectral Sciences (http://modtran.spectral. com, licence required, last access: TS8).

*Data availability.* The surface models are either default MODTRAN or available from the Isofit GitHub under data/reflectance/surface_model_ucsb TS9. The instrument noise model is from the Isofit GitHub under data/sbg_noise_coeffs.txt(https://doi.org/10.5281/zenodo.4614338, Brodrick et al., 2021 TS10). The LES output was generated using published large eddy simulation models, and the cases are described in the references in row (xi) of Table 1.

*Supplement.* The supplement related to this article is available online at: https://doi.org/10.5194/amt-14-1-2021-supplement.

*Author contributions.* The authors contributed as follows: MTR (conceptualisation, investigation, methodology, visualisation, writing), DRT (methodology, writing and resources, namely Isofit), MJK (writing and resources, namely LES output), and MDL (conceptualisation, methodology, writing).

*Competing interests.* The authors declare that they have no conflict of interest.

*Acknowledgements.* The research described in this paper was carried out at the Jet Propulsion Laboratory, California Institute of Technology, under contract with the National Aeronautics and Space Administration. The authors also thank Amin Nehrir and Brian Carroll for assistance with the HALO lidar data.

*Financial support.* This research has been supported by the National Aeronautics and Space Administration (grant no. NNN12AA01C). TS11

*Review statement.* This paper was edited by Alexander Kokhanovsky and reviewed by two anonymous referees.

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

## Remarks from the typesetter

TS1  Please confirm.

TS2  Please give an explanation of why this needs to be changed. We have to ask the handling editor for approval. Thanks.

TS3  Please confirm date.

TS4  Please give an explanation of why this needs to be changed. We have to ask the handling editor for approval. Thanks.

TS5  Please give an explanation of why this needs to be changed. We have to ask the handling editor for approval. Thanks.

TS6  Please confirm removal of the reference.

TS7  Please confirm date, added DOI and citation.

TS8  Please provide date of last access and the reference list entry.

TS9  Please confirm.

TS10  Please confirm added DOI and citation.

TS11  Please note that the funding information has been added to this paper. Please check if it is correct. Please also double-check your acknowledgements to see whether repeated information can be removed or changed accordingly. Thanks.

TS12  Please confirm names and initials.

TS13  Please confirm added information.

TS14  Please confirm article number.

TS15  Please confirm author name.

TS16  Please confirm publisher and place of publication.