# Peer review of "Boundary layer water vapour statistics from high-spatial-resolution spaceborne imaging spectroscopy"

_Atmospheric Measurement Techniques, 2021_

## Author Response (AR1)

Dear Editor,

Thanks for taking on our paper, we have now addressed all review comments. This file contains most of the same content as the public comment responses.

A major issue the reviewers raised was a request for "real" data. We have added some statistics from published AVIRIS-NG flights, and list a number of other published results, but now explain with additional Section 3.3 text how and why our main results cannot currently be validated. In Section 5 we comment on how this could be addressed in potential airborne campaigns.

This file contains Section 1, which describes changes we made that weren't in response to reviewer comments, none of which change our results. Section 2 responds to review 1 and Section 3 to review comment 2. Reviewers are in black text, our responses in red, and we start each new section on a new page.

**1. Joint Response**

Please note that we found and corrected some figure, table and notation errors, these do not affect results.

Figure 5 – removed y ticklabels from (c,f). Added axis labels to edge subplots.

Figure 9 – x-axis label added to panel (c).

Figure 11 – axis labels added.

Table 1 – we accidentally included an old version calculated from bugged code, and have now corrected rows (vii)—(ix). The LES vertical grid definitions differ between simulations (either Arakawa A-grid or C-grid). The biggest issue was that for C-grids our early code smeared upper-LES $q_v$ into the reanalysis layer above, resulting in way too much TCWV. This was fixed before submission *except* in Table 1. For example, RICO TCWV was originally 49.6—49.7 mm in Table 1, but the new ~37 mm agrees with Figure 8. The BOMEX LES only includes ~86 % of the total TCWV so P4L17 has been changed from "…show that the LES capture >90 % of total TCWV…" to "…show that the LES capture >85 % of total TCWV…".

Equation (4) had a mix of $r$ and $\Delta r$, we changed to consistent use of $\Delta r$ to emphasise it's a gap between points.

We also had feedback that the error estimation method in Sec. 3.1.4 could be confusing. To our knowledge it is new, and while we are very happy with its performance it isn't completely intuitive. We have rephrased some text in Section 3.1.4 and referred to new Supplementary Figures 9—11. These figures show a step-by-step guide to our method, and compare results with a more standard/intuitive method based on spatial filters.

**2. Reviewer 1**

I congratulate the authors for a good study and a good statistical work.

Thanks for your generous comments and thoughtful review. We also apologise for not responding during the open review period to allow time for a continuous discussion – we simply weren't able to do everything in time. We believe we have addressed all of your comments, however.

I have some minor comments:

* general:

- I miss a lack of comparison with real data. The spatial uncertainties resulting from this study could have been compared with real data under similar conditions as the simulations (e.g. flat-terrain, etc..)

We have added extra text: the abstract now introduces this as an observing system simulation experiment, the introduction makes it clear that this is primarily a model sensitivity study and Section 3.3 adds several paragraphs with comparison numbers but also explains why we can't yet observationally evaluate our $\sigma_x$ estimates.

Section 3.3 shows our results are consistent with RMSE reported for satellite-surface station comparisons for other studies. We cite OCO-2, MODIS, MERIS, OLCI and Sentinel-2 as examples. We also report some AVIRIS-NG flight statistics, but mention that those also aren't directly comparable because it's a different instrument and those are flights over sunglint ocean.

We also now describe in Section 3.3 and 5 how future measurements, perhaps from airborne campaigns with collocated data, could allow us to do better evaluations of our target statistics, and most importantly $\sigma_x$.

We could have grabbed some TCWV retrieval fields from other sensors, but their retrievals might be very different, they might have different sensitivities to surface type etc, and that might require further detailed investigation to understand. The AVIRIS-NG numbers are the most consistent with ours, so we just report those as an example.

- In such detailed statistical work I miss also an estimation of the order of magnitude of the contribution of the different approximations in the model to the final results.

We struggled to interpret this comment but we think that the current manuscript addresses the main relevant issues. We explain that we can remove random error and identify the main source of error in $\sigma_x$ as likely due to retrieval assumptions about the atmospheric profile but don't believe we can go further.

Some of this is related to the above point: we lack independent validation data.

1) We cannot include some errors, like spectroscopy or quantify RT assumption errors with this setup. This would require independent data.
2) We have added a prior sensitivity test, showing <0.15 mm mean shift with no effect on $\sigma_x$ (Section 3.1.1 new text, Supplementary Figure 3).
3) We show that random retrieval error can be removed almost perfectly (Figure 7)
4) We say in Figure 7 discussion and later that the biggest uncertainty source relevant for our $\sigma_x$ is the gradient $dTCWV_{ret}/dTCWV$, which we propose may be driven by errors in the atmospheric profile.

Re-reading, we think that *for our main results* (i.e. spatial statistics) we make it clear that the gradient a1 parameter is the most important. A relevant updated Section 5 sentence reads:

"In Isofit, the atmospheric component contributes a bias to $dTCWV_{ret}/dTCWV$ and may be the largest source of our errors in $\sigma_x$, which range from -7 % to +34 % of true $\sigma_x$."

\* page 1:

- "... upcoming missions such as the Earth Surface Mineral Dust Source Investigation (EMIT) will offer unprecedented horizontal resolutions of order 30 - 80 m..." -> currently there are several missions (VNIR - VSWIR) with this or even better resolution (e.g. multispectral Sentinel-2 (20 m) and hyperspectral DESIS) with bands in the water vapor absorption regions. Actually, wouldn't it be more sound to make this study with the Sentinel-2 20m resolution?.

We now realise that our submitted intro was confusing since we mixed up discussion of our hyperspectral work with later reference to multispectral results from MERIS. We have fixed this by rephrasing throughout; we now use phrasing similar to "modern and upcoming" in the abstract and all sections. We have added a paragraph to Section 1 discussing PRISMA, DESIS and Sentinel-2.

We also added the following justification to the intro and hope this efficiently justifies our choice:

"The purpose of this is a detailed sensitivity study using retrieval code and tools already developed for EMIT. We consider $\Delta x \geq 40$ m since this is appropriate for EMIT and several LES cases in our archive that were run at that resolution."

We didn't make a big deal about it, but Figure 9 shows the $\Delta x=20$ m results for the two simulations that were run at that resolution. We argue that the use of a retrieval code that will be used for sensors at this resolution + the availability of LES simulations justifies out choice here. Finer resolution work would of course be welcome & interesting to us!

The study would also profit of the large amount of real data, which leads me to the first general comment.

We agree, obviously! Please see changes to Section 3.3 and argument as to why the available data are not directly comparable for our results.

\* page 7:

- is a plane-parallel atmosphere still a good approximation for SZA = 45?. The effect in sigmax seems to be of the order of 0.025 (figure 12), the same as the difference between 50 - 300 m resolution. This could be included in the second general comment.

This is a tricky point – we only have plane-parallel RT output.

We don't think the Figure 12 results should be very sensitive to this though, since this atmosphere isn't strongly scattering except for potential influence from cloud 3D radiative effects. This is a really complex problem that needs different (expensive!) radiative transfer tools to address. We have discussed doing this if future time and funding permits, but it would really need a whole additional study.

Successful error budget closure in past AVIRIS Isofit work gives us some confidence that non-plane-parallel effects aren't too important, but we have added text to draw some extra attention to this, e.g. Section 3.1 and Section 5 (added text in *italic*):

Section 3.1 : "We first remind readers that "retrieval error" here only includes errors present in these synthetic retrievals, and excludes several real-world sources, such as how the true atmosphere is not plane-parallel as assume in our radiative transfer"

Section 5: "Future work could address uncertainties that are ignored here, such as topography or cloud 3-D radiative effects *via 3-D radiative transfer simulations which avoid several of our assumptions, such as a plane-parallel atmosphere*"

- I did not find the Supplementary figures

Apologies for the confusion, these were uploaded as a separate file at https://amt.copernicus.org/preprints/amt-2021-89/amt-2021-89-supplement.pdf

Our review response files will include an updated SI to include a new table on SZA change effects and on a TCWV prior sensitivity effect.

\* Page 10:

- The difference between different surface brightness seems to be of the same order of magnitude of the dispersion within the same surface brightness. And there is a much larger offset between the retrieved TCWV and the true one. The offset seems to be smaller for for brighter surfaces, but there is still 1mm difference for 50% retrieved surface reflectance.

This is one of many results we clipped to stop the paper from getting even longer. In hindsight it deserves comment so we have added the following text to the Section 3.2.1 paragraphs discussing Figure 4:

"Regardless of the surface, a bias of order ~1 mm remains, which is similar to the largest difference introduced by surface type and may be related to other retrieval errors such as inappropriate atmospheric profile shapes assumed in the LUT. However, the derived spatial statistics we are interested in here are not affected by any mean bias."

This is another handy reminder that we are interested in $\sigma_x$, not absolute biases here, and so feeds into our new Section 3.3 discussion on observational comparisons.

\* Page 15:

- which is the typical error of Isofit with respect to WV true measurements?

The AERONET results from Thompson et al. (2021) are now mentioned in the new Section 3.3.

- which depend the TCWV variability of -7% to 34% of?. E.g. Is it a function of the true TCWV?

We have added "of true $\sigma_x$".

This comes from our gradient calculation. Running through the maths it's not dependent on true TCWV but simple scales the standard deviation.

- I have missed some conclusions for smaller spatial resolutions sensors.

We are not sure how to interpret this comment. We *think* we have addressed it with our added text.

Section 1: introduces Sentinel-2 and explains why we're not doing 20 m.

Section 3.3: discusses errors

Section 5: Added text:

"Finally, this work could be extended to other sensors, such as MSI on Sentinel-2, which is not hyperspectral but provides exceptionally fine $Dx$ of approximately 20 m. Additional high-resolution analysis may be required for this, since **Error! Reference source not found.**(a,b) imply increases in retrieved $\sigma_x$ at $\Delta x$=20 m for the two simulations that were run at that resolution."

If some of my comments are already explained somewhere in the text, I would thank the authors to point me to the section containing the explanations.

Unfortunately I did not have the chance to read the article in a row and I might have missed some of the explanations to my comments.

Once again, we appreciate your efforts for this review and understand going through this paper piecemeal. We had to address a *lot* of potential issues since this is intended to lay the groundwork for a lot of future analysis.

**3. Reviewer 2**

The manuscript presents new retrieval statistics for planetary boundary layer (PBL) water vapor from high-spatial resolution spaceborne imaging spectroscopy. The authors focus on a sensitivity analysis based on a coupled forward and inverse modeling in the frame of the Earth Surface Mineral Dust Source Investigation (EMIT) mission. They analyze uncertainties introduced by instrument errors, surface type, and varying solar zenith angle (SZA), and assess the overall potential of upcoming spaceborne high-resolution VSWIR instruments.

The study is really interesting, novel and well written. Especially the presentation of the statistical evaluation is of high quality. If proven robust also for real data, the concept will be of great interest for the atmospheric science community. However, I have a few comments, which follow below.

Thanks for the positive feedback and for your attentive reading. We genuinely appreciate your efforts to follow and critically evaluate this lengthy paper. We tried to balance readability and relevance without going even longer but your perspective has helped us to re-evaluate our choices.

Regarding use of "real" data, we now clearly state in the abstract that this is an observing system simulation experiment, and in the intro that this is a model sensitivity study, we then discuss in more detail the difficulty of observational comparisons (Section 3.3) and mention how this might be tested in future airborne campaigns or be thought about for other missions (Section 5).

General comments

- The introduction could benefit from a clearer structure. While the two closing paragraphs including the four research questions are distinct and coherent, the remaining part could be more explicitly separated into literature research describing previous work and theoretical concepts on the one hand, and presenting the novelty and methodologies applied in this study on the other. For instance, lines 7-11 could be moved to the final paragraphs of the introduction combined with a little rephrasing.

We have restructured:

1. Decadal survey detail added early on
2. Discussion of DESIS, PRISMA, MSI on Sentinel-2 added as new paragraph
3. The pg2 lines 7—11 text has been modified and moved to just before the final two paragraphs of the introduction.
4. It's now specified that this is a sensitivity study targeted at EMIT, although many of the principles should apply to other VSWIR instruments.

- The discussion part of Section 3 could be improved by a more detailed comparison with retrieval results from already existing instruments such as MERIS. You are listing several instruments for TCWV retrievals in the introduction and the reader could get a better impression of the retrieval performance from synthetic EMIT data in case some reference values from other datasets are given.

We have added several paragraphs to Section 3.3. We try to emphasise the following points:

(i)      Observational study RMSE includes our bias+spatial variability+retrieval error+differences between LES cases. Our bias+spread is right ball park compared with these other instruments.

(ii)     Previously published AVIRIS-NG Isofit results are also consistent, and the derived spatial variability and random error from one of the Thompson et al. (2021) flights is within the range we explore in this paper.

- Overall, the manuscript would strongly benefit from an application of the presented methodology to "real" data. You basically agree with this on page 12, lines 26-28, with the statement "Limitations include the use of the same radiative transfer code for forward and inverse simulations…". For instance, the Italian PRISMA instrument already delivers high-spatial resolution imaging spectroscopy data and could be used for PBL TCWV retrievals in the same manner as EMIT. On the other hand, if this study is intended to serve as a pure sensitivity analysis, this should be clearly mentioned in the introduction.

In the new Section 3.3 we discuss how standard comparisons don't tell us much about our retrieved $\sigma_x$, since we would need independent measurements of TCWV that are exactly collocated at the same resolution.

We checked the PRISMA data here: http://spazio-news.it/asi-prisma-da-oggi-la-comunita-scientifica-puo-accedere-ai-dati

l'Agenzia Spaziale Italiana wants a detailed licence agreement with description of data use etc. Given the limitations of any comparison that we mention in Section 3.3, we're unsure of getting anything useful out of such a comparison.

We think the new Section 3.3 text justifies this choice and we added extra text in Section 5 to help point out future ways to address this limitation.

Specific comments

Page 1, lines 24-27: Could you provide a little bit more context why the knowledge about vertical moisture structure of the atmosphere is crucial for weather and climate applications, and why thermodynamic information is a targeted observable recommended by NASA's Decadal Survey?

We have added a quotation from the Decadal Survey and touched on what we identify as the three main reasons: PBL-surface coupling, PBL-troposphere coupling, and within-PBL cloud formation. We also only state that we are filling in just one measurement gap, and not fully addressing the goals of the survey, which also include more vertical detail and diurnal sampling. That's a rabbithole we avoid since it adds little to our study.

Page 2, line 14: "However, similar capacity is anticipated…". I would go beyond and replace "similar" with "improved" since SBG and CHIME will most likely be offering even higher spatial-resolution than EMIT.

We changed this to "similar or superior" since we do not know the full specs and in-orbit performance yet, and "capacity" could be interpreted in different ways, e.g. swath size, sampling/revisit time, instrumental noise etc.

Page 3, lines 12-14: The reader could get the impression that the AVIRIS-NG flights were selected for this study. Please try to rephrase and clarify.

Changed to: "Thompson et al. selected these flights…"

Page 3, line 18: Although it is explained later on, it would be good to have a short definition of the "true TCWV" here, e.g., "…, which was used as input for our forward simulations…".

Change made

Page 6, line 8: Please define the quantity $\rho_s$.

"surface reflectance" inserted before $\rho_s$.

Page 6, line 10: Don't you miss to list the spherical sky albedo here when mentioning the flux calculations coming from MODTRAN?

This is a good catch, thanks. We changed the ordering of this section a few times before submission and have carefully re-read it. We think all properties are now properly introduced.

Page 6, line 11: I think it would be better to say "spectral response function (SRF)" instead of "line shape (ILS)". This might be more common in the remote sensing community.

Change made, but we don't refer back to it so acronym removed.

Page 6, lines 13-15: You could add references to Rothman et al. (2009) for HITRAN and to Stamnes et al. (1988) for DISORT here.

Citations added, and "for summary see…" removed.

Page 6, lines 14-15: Is the number of DISORT streams of importance for your application? Either remove it or explain why you used 8 streams.

This was inserted on autopilot by someone who's spent a lot of time doing RT sensitivity tests. This matches the default Isofit configuration and isn't important for our application, so we removed it.

Page 6, lines 24-25: You define the used reflectance quantity as the hemispheric-directional distribution function on page 7. However, it would be good to have the definition here, directly after introducing Eq. (1).

The HDRF mention and Schaepman-Strub reference have been cut and pasted directly after the reflectance sentence following Eq. (1).

Page 7, lines 4-6: Did you normalize the surface prior distribution to avoid constraints on the reflectance magnitude as described in Thompson et al. (2018)?

We did not – this allows us to show absolute magnitude of the surface spectra in Figure 4. We have inserted:

"We retrieve absolute $\rho_s$, rather than the normalised value discussed in Thompson et al. (2018), and the…"

Page 7, line 16: Which performance do you mean here? Give quantities.

Oops – the SI text next to (original) Supplementary Figure 3 (now 4) has "Error! Reference not found". There was supposed to be a Supplementary Table. We have moved this part of the text after Eq. (2) which describes the emulator, so we can describe performance in terms of emulator parameters:

"Tests with SZA from 14—60° show no significant differences in $a_1$ with SZA, while the standard deviation of $\epsilon$ increases by up to 25 % at SZA=60° relative to SZA=45° (Supplementary Figure 5, Supplementary Table 1). Section 3.1.4 shows how we are able to identify and remove the effect of $\epsilon$ on derived statistics, so we anticipate that our conclusions will largely apply to SZA up to 60°."

The Supplementary Figures have been re-ordered to match their introduction in the text following the movement of this sentence.

Page 8, lines 1-2: What about other types of surfaces such as artificial surfaces or snow? Do you plan to extend your analysis to those types as part of future work? If yes, this could be mentioned in the discussion/conclusion of your results.

We felt it make sense to address the artificial surfaces at this point and have added: "The database used to generate the surface model includes artificial surfaces, which are captured in the "mineral" spectra." This is the default surface model distributed with isofit from github, we think that just mentioning artificial materials' inclusion is sufficient.

In Section 5 we have added a paragraph on how current development will add snow surfaces, and a quick comment on how its spectral shape might matter.

Page 10, line 8: What does "retrieved well" mean? Give quantities.

We have added:

"Surface $\rho_s$ are retrieved well, with mean bias magnitude equivalent to 0.3—1.6 % of true $\rho_s$ (e.g. for Lambertian $\rho_s$=0.1, the mean bias is 0.00021) and standard deviation of 2—4 % of true $\rho_s$."

It's quite fiddly phrasing but we think this provides the necessary information. We calculated values from the Lambertian surface errors because for the other surface spectra there would be ambiguity between (1) calculating percentage at each wavelength, then mean of those and (2) calculating the mean bias in $\rho_s$, and then turning that into a percentage.

Page 10, line 12: Which prior mean and covariance did you use for the TCWV state vector parameter in your ISOFIT setup? And did you use the default first guess estimation based on a heuristic band ratio retrieval? It would be good to provide this information earlier in Section 3.1.1 and to discuss it in a few words as it can influence your retrieval results.

A paragraph has been added to the end of Section 3.1.1. It's 40.0±7.5 mm in all cases but we originally cut our early sensitivity tests for length. We now show one of them in Supplementary Figure 3, in which we pick an extremely low prior of 7.5±7.5 mm. The effect on retrieved TCWV is minor: no change in gradient so $\sigma_x$ is unaffected. A 0.15 mm change in bias, which is ~15 % of total bias.

[Figure]

Technical corrections

We appreciate your painstaking reading and have made the suggested corrections, except where noted.

Page 2, line 10: Although "LES" is defined in the abstract, it would be nice to have the full expression here again.

Done.

Page 2, line 26: Rephrase to "…via two demonstrated approaches **in order to** provide a single value…".

We rephrased the sentence instead in order to avoid repetition.

Page 3, lines 4-5: "More TCWV **leads to increasing depth of $H_2O$ absorption features** relative to other wavelengths."

Done.

Page 3, line 10: "**The** retrievals…".

Done.

Page 6, line 8: Rephrase to "Conceptually**,** it targets $\rho_s$ and **the estimation of** TCWV is seen as part of an atmospheric correction."

Done.

Page 6, line 24: "…**the cosine of** the solar zenith angle,…"

Done.

Page 6, line 31: "…**generated** using…"

Done.

References

**Rothman**, L.S., Gordon, I.E., Barbe, A., Brenner, D.C., Bernath, P.F., Birk, M., Boudon, V.,Brown, L.R., Campargue, A., Champion, J.P., Chance, K., Coudert, L.H., Diana, V., Devi, V.M., Fally, S., Flaud, J.M., Gamache, R.R., Goldman, A., Jacquemart, D., Kleiner, I., Lacome, N., Lafferty, W.J., Mandin, J.Y., Massie, S.T., Mikhailenko, S.N., Miller, C.E., Moazzen-Ahmadi, N., Naumenko, O.V., Nikitin, A.V., Orphal, J., Perevalov, V.I., Perrin, A., Predoi-Cross, A., Rinsland, C.P., Rotger, M., Simeckova, M., Smith, M.A.H., Sung, K., Tashkun, S.A., Tennyson, J., Toth, R.A., Vandaele, A.C., Auwera, J.V., 2009. The HITRAN 2008 molecular spectroscopic database. *J. Quant. Spectrosc. Ra.* 110, 533–572. doi:DOI: 10.1016/j.jqsrt.2009.02.013.

**Stamnes**, K., Tsay, S.C., Wiscombe, W., Jayaweera, K., 1988. A numerically stable algorithm for discrete ordinates method radiative transfer in multiple scattering and emitting layered media. *Appl. Optics* 27, 2502–2509.

**Thompson**, D.R., Natraj, V., Green, R.O., Helmlinger, M.C., Gao, B.C., Eastwood, M.L., 2018. Optimal estimation for imaging spectrometer atmospheric correction. *Remote Sens. Environ.* 216, 355–373. doi:10.1016/ j.rse.2018.07.003.